

# Inferring and evaluating satellite-based constraints on NOx emissions estimates in air quality simulations

James D. East[1,2], Barron H. Henderson[3], Sergey L. Napelenok[3], Shannon N. Koplitz[3], Golam Sarwar[3], Robert Gilliam[3], Allen Lenzen[4], Daniel Q. Tong[5], R. Bradley Pierce[4], Fernando Garcia-Menendez[1]

[1]Department of Civil, Construction, and Environmental Engineering, North Carolina State University, Raleigh, NC 27695, USA
[2]Oak Ridge Institute for Science and Education, Office of Research and Development, U.S. Environmental Protection Agency, Research Triangle Park, NC 27711, USA
[3]U.S. Environmental Protection Agency, Research Triangle Park, NC 27711, USA
[4]Space Science and Engineering Center, University of Wisconsin-Madison, Madison, WI 53706, USA
[5]Department of Atmospheric, Oceanic and Earth Sciences, George Mason University, Fairfax, VA 22030 USA

*Correspondence to*: Fernando Garcia-Menendez (f_garcia@ncsu.edu)

**Abstract.** Satellite observations of tropospheric $NO_2$ columns can provide top-down observational constraints on emissions estimates of nitrogen oxides ($NO_x$). Mass-balance based methods are often applied for this purpose, but do not isolate near-surface emissions from those aloft, such as lightning emissions. Here, we introduce an inverse modeling framework that couples satellite chemical data assimilation to a chemical transport model and infers satellite-constrained emissions totals using the iterative finite-difference mass-balance method. The approach improves the finite-difference mass-balance inversion by isolating the near-surface emissions increment. We apply the framework to estimate lightning and anthropogenic $NO_x$ emissions over the Northern Hemisphere. Using overlapping observations from the Ozone Monitoring Instrument (OMI) and the Tropospheric Monitoring Instrument (TROPOMI), we compare $NO_x$ emissions inferences from these satellite instruments, as well as the impacts of emissions changes on modeled $NO_2$ and $O_3$. OMI inferences of anthropogenic emissions consistently lead to larger emissions than TROPOMI inferences, attributed to a low bias in TROPOMI $NO_2$ retrievals. Updated lightning $NO_x$ emissions from either satellite improve the chemical transport model's low tropospheric $O_3$ bias. Combined lightning and anthropogenic updates inferred from satellite observations can improve the model's ability to represent background and ground-level $O_3$ concentrations, an ongoing policy consideration in the U.S. as domestic and international emissions control strategies evolve.

## 1 Introduction

Tropospheric nitrogen oxides ($NO_x$), nitric oxide (NO) and nitrogen dioxide ($NO_2$), harm human health (Anenberg et al., 2018; Murray et al., 2020) and play a key role in the formation of important secondary atmospheric pollutants, such as $O_3$ (Jacob, 2000). $NO_x$ is emitted to the troposphere primarily by anthropogenic combustion processes, but natural sources, including soil, lightning, and wildfires, also contribute to the atmospheric $NO_x$ budget (Jacob, 1999). Accurate $NO_x$





emissions are a critical component of local- to global-scale atmospheric chemistry simulations. On hemispheric scales, realistically representing the formation and intercontinental transport of $O_3$ with models requires adequate international emissions inventories (Itahashi et al., 2020; Zhang et al., 2016; Zhang et al., 2008; Verstraeten et al., 2015; Mathur et al.,

2017). In regional air quality simulations, which commonly rely on hemispheric or global models for chemical boundary conditions, the relative contribution of long-range pollutant transport to ground-level $O_3$ concentrations has grown in many areas as $O_3$ precursor emissions have decreased in the U.S. and other developed countries (McDuffie et al., 2020; Jaffe et al., 2018; Simon et al., 2015). As a result, air quality management policies, often informed by regional modeling, are strengthened by accurate and up-to-date global $NO_x$ emissions inventories. However, compilation of bottom-up regional and

global emissions inventories, developed from source- and location-specific emissions factors and activity data, is time- and labor-intensive, and can be hindered by limited data. As a result, bottom-up inventories lag emissions changes and are often incomplete. Although uncertainties in bottom-up emissions estimates are particularly large for developing countries (McDuffie et al., 2020; Elguindi et al., 2020), they remain significant for developed countries as well (Day et al., 2019).

      Satellite observations of $NO_2$ can bridge temporal gaps in emissions estimates (Tong et al., 2016; Tong et al., 2015) and

constrain uncertainty in emissions inventories through inverse modeling (e.g. Lamsal et al., 2011; Goldberg et al., 2021; de Foy and Schauer, 2022). Several methods have been applied to develop top-down emissions estimates using satellite observations and atmospheric models, each carrying advantages and limitations (Elguindi et al., 2020). Adjoint-based methods can provide detailed emissions updates, but require significant computational resources (e.g. Qu et al., 2017; Qu et al., 2019; Muller and Stavrakou, 2005; Kurokawa et al., 2009; Cooper et al., 2017; Zhang et al., 2019; Wang et al., 2020b).

Similarly, Kalman filtering and related approaches have been used but are computationally-intensive (e.g. Napelenok et al., 2008; Ding et al., 2020; Ding et al., 2015; Mijling and Van Der A, 2012; Miyazaki and Eskes, 2013; Miyazaki et al., 2017; Miyazaki et al., 2012a; Miyazaki et al., 2012b; Sekiya et al., 2021). Mass balance inversion approaches, which scale model emissions by directly comparing model estimates and satellite observations, were introduced by Martin et al. (2003), updated by Lamsal et al. (2011), and have been widely used in research and forecasting (e.g. Boersma et al., 2015; Itahashi et al.,

2019; Li et al., 2018; Visser et al., 2019; Zhu et al., 2021; Cooper et al., 2017). Although lower computational costs allow the finite-difference mass-balance (FDMB) approach (Lamsal et al., 2011) to readily update emissions, the method is subject to an emissions smearing effect (Cooper et al., 2017). Since FDMB uses satellite observations directly, near-surface $NO_2$ bias cannot be isolated from biases in the middle and upper troposphere, which obscures the surface emissions inference. Further, applications often rely on a single inversion from a single satellite, although available satellite products have been

shown to have significant biases. For example, early versions the Tropospheric Monitoring Instrument (TROPOMI) $NO_2$ product showed a low bias in urban areas when compared against surface-based and airborne spectrometer measurements (Judd et al., 2020; Verhoelst et al., 2021) and the Ozone Monitoring Instrument (OMI) $NO_2$ product has been reported to differ with measurements by $\pm 20\%$ (Lamsal et al., 2014). The impact of biases in satellite-based $NO_2$ data on mass-balance inversions has not been fully explored despite the wide used of the method to scale $NO_x$ emissions. Minimizing bias in





anthropogenic emissions inferences and understanding the potential for them to propagate to emissions updates are needed to improve mass-balance-based inversions.

Here, we introduce a modeling framework that couples satellite chemical data assimilation to the Community Multiscale Air Quality model (CMAQ) and applies an iterative FDMB inversion to estimate $NO_x$ emissions in the Northern Hemisphere. The framework provides observational constraints to improve emissions estimates in areas where emissions are
highly uncertain, at a lower computational cost relative to adjoint- and Kalman-filter-based approaches. We apply the framework in an iterative assimilation to infer 2019 $NO_x$ emissions, the first complete year in which OMI and TROPOMI records overlap. In contrast to traditional FDMB, which directly compares modeled and observed columns, our framework improves the FDMB method by first assimilating satellite-retrieved $NO_2$. Assimilating the observed column allows the inversion to target near-surface $NO_2$ and minimize influences from the upper troposphere, extending the framework
proposed by Lamsal et al. (2011). In addition, our analysis compares inversions using OMI and TROPOMI $NO_2$ data. We show that the inverse emissions produced by this framework influence representation of intercontinental $O_3$ transport to the U.S., offering an opportunity to improve chemical boundary conditions in policy-relevant regional-scale air quality simulations.

## 2 Methods

We develop a framework to update $NO_x$ emissions estimates using the CMAQ chemical transport air quality model (Byun and Schere, 2006), 3D-variational (3DVAR) chemical data assimilation (Sandu and Chai, 2011), and space-based $NO_2$ observations. We apply the framework to estimate 2019 lightning and anthropogenic $NO_x$ emissions, and compare surface- and space-based $NO_2$ observations to model simulations using the prior emission (inventory before the framework is applied) and posterior emissions (inventory after the framework is applied) to assess the impact of the updates. Figure 1
provides an overview of the framework, in which lightning $NO_x$ ($LNO_x$) emissions and anthropogenic $NO_x$ ($ANO_x$) emissions are updated separately.





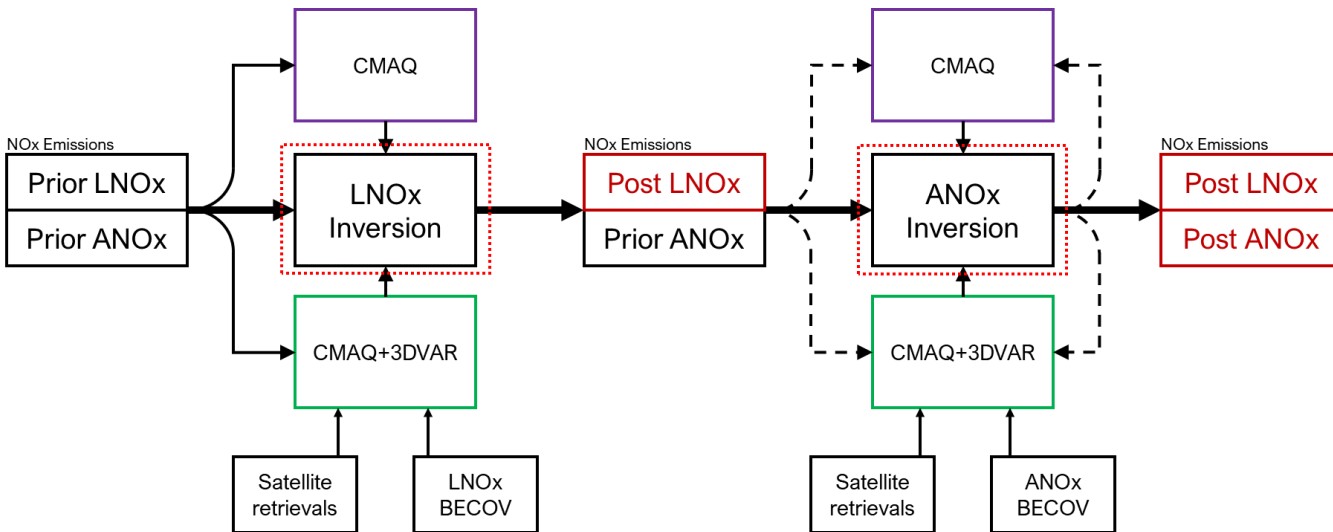

**Figure 1:** NO$_x$ emissions inversion framework. Lightning NO$_x$ (LNO$_x$) emissions are updated in the first step. Then, anthropogenic NO$_x$ (ANO$_x$) emissions are updated iteratively. CMAQ boxes represent air quality simulations without chemical data assimilation, and CMAQ+3DVAR boxes represents air quality model simulations with chemical data assimilation. Satellite NO$_2$ retrievals and a background error covariance (BECOV) are inputs to the chemical data assimilation, described in Sect. 2.3. Red dotted lines around the inversion boxes represent the boundary of the inversion algorithm, which is detailed in Fig. 3. Dashed black emissions input lines around the ANO$_x$ inversion represent the iterative process. Iteration and convergence criteria are described in Sect. 2.6.

### 2.1 Satellite data

We use NO$_2$ tropospheric column observations from the National Aeronautics and Space Administration's (NASA's) OMI and the Royal Netherlands Meteorological Institute's (KNMI's) TROPOMI instruments in the inversion framework. TROPOMI was launched in October 2017 and provides 7.2×3.6 km$^2$ resolution NO$_2$ retrievals, upgraded to 5.6×3.6 km$^2$ resolution in August 2019 (Van Geffen et al., 2020; Veefkind et al., 2012). TROPOMI's sun-synchronous polar orbit crosses the equator at approximately 1:30 pm local time, allowing the instrument to achieve global coverage in one day. We assimilate the Level-2 tropospheric slant column retrieved from NASA's Earth Science Data Systems Program (https://earthdata.nasa.gov/). The data product is described in the Algorithm and Theoretical Basis Document (ATBD) for TROPOMI NO$_2$ (Van Geffen et al., 2019). We only consider TROPOMI observations with a quality flag greater than 0.5 and cloud fraction lower than 30% in the assimilation, following data product recommendations (Eskes et al., 2019). We use the latest publicly available versions of the TROPOMI retrieval for 2019 (versions 1.2.2 to 1.3.2) at the time of the analysis. Version 1.3 introduced updates to cloud processing that decrease noisy hotspots and broadened the range of acceptable air mass factors (Eskes et al., 2021). Information about the updates applied in each version and the dates on which updates were applied is given in Eskes et al. (2021). A research version with an updated retrieval applied to 2019 observations has been developed (Van Geffen et al., 2021), but was not yet standard and was not available at the time of this analysis. We discuss the impact of these latest updates in Sect. 3.3.





OMI, onboard the Aura satellite launched in 2004, provides tropospheric $NO_2$ vertical and slant column retrievals with a resolution of $13 \times 24$ km$^2$ near nadir in a sun-synchronous polar orbit with a local equator crossing time of 1:45 pm. Global coverage is achieved in two days. We use the NASA Goddard Space Flight Center (GSFC) Level-2 $NO_2$ product (Krotkov et al., 2019b). OMI was impacted by a row anomaly beginning in 2008, reducing the number of usable pixels in the OMI retrieval (Boersma et al., 2018). We include only pixels with cloud fraction lower than 30% and a summary quality flag of 0.

Detailed information about the $NO_2$ data product is included in the OMI ATBD (Chance, 2002) and in Krotkov et al. (2019a).

A low bias has been noted in the versions of TROPOMI $NO_2$ used for this study (Judd et al., 2020; Verhoelst et al., 2021). Although TROPOMI $NO_2$ in 2019 has been reprocessed with retrieval version 2.3.1, resulting in an improvement of the bias (Eskes et al., 2021), these reprocessed datasets were not yet available at the time this analysis was conducted. Figure

2 compares TROPOMI and OMI tropospheric vertical column density (VCD) for 2019, regridded to the CMAQ grid used. For the VCDs shown in the figure, we remove the effect of the assumed vertical profile of $NO_2$ from the original satellite product by recalculating the VCDs with the $NO_2$ vertical profile simulated by CMAQ. In the Results, we discuss the low bias in TROPOMI data and explore its impact on emissions inversions.

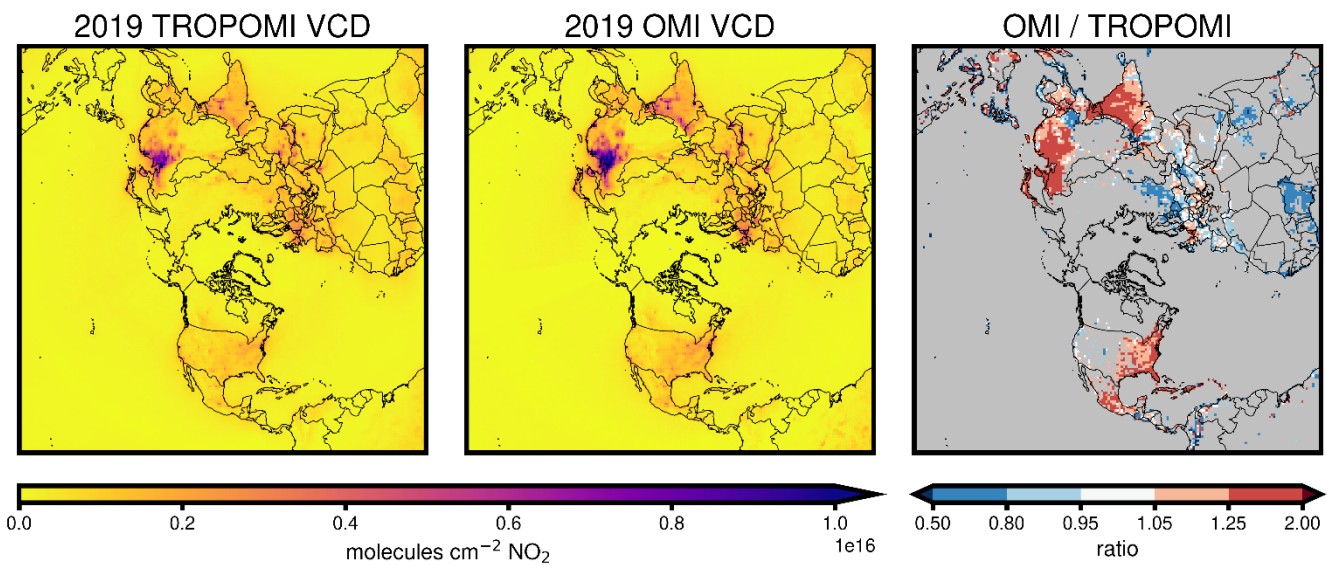


**Figure 2:** 2019 annual average TROPOMI and OMI vertical $NO_2$ vertical column densities, with CMAQ $NO_2$ profiles applied, and the ratio between them. Column densities ratios are only shown for the grid cells where $NO_x$ emissions updates are applied in the emissions inversion.



## 2.2 Hemispheric air quality modeling

Model simulations in the inversion framework were completed for January–December 2019 using CMAQ v5.3.2 (Appel et al., 2021; U.S. EPA, 2020). CMAQ has been used to simulated air quality over the Northern Hemisphere and shown to adequately capture chemical composition against observations (Mathur et al., 2017). Model inputs and satellite observations are summarized in Table 1. Simulations, designed to capture continental-scale pollutant transport, cover the Northern Hemisphere with 108 km horizontal grid spacing and a 44-layer vertical structure reaching 50 hPa (Mathur et al., 2017). The simulations use version CB6r3 of the Carbon Bond 6 chemical mechanism (Luecken et al., 2019), the AERO7 aerosol module (Xu et al., 2018), and updated halogen chemistry (Kang et al., 2021). Anthropogenic emissions are modeled using representative day-of-week emissions that change month-to-month. Representative-day emissions are created by averaging data from the prior emissions inventory on a day-of-week basis by month. The prior emissions inventory, relies on the best available emissions data at the time of the study. Anthropogenic emissions for North America are from the U.S. Environmental Protection Agency's (EPA) 2017 National Emissions Inventory (NEI) modeling platform (Adams, 2020). Emissions in China are for the year 2015 (Zhao et al., 2018) and emissions for the rest of the hemisphere are based on the Hemispheric Transport of Air Pollution (HTAP) version 2, projected from their original 2010 date to 2014 with scaling factors from the Community Emissions Data System (CEDS). To initialize the 2019 prior and posterior simulations and reduce the impact of chemical initial conditions on the results, we use a 1-year spin-up period not considered for the analyses. CMAQ model runs are driven by meteorology from a retrospective hemispheric simulation using the Weather Research and Forecasting (WRF) model (Skamarock et al., 2008) version 4.1.1 configured following Mathur et al. (2017) and Xing et al. (2015).

**Table 1:** Prior emissions and model inputs

| Data | Year | Source |
| --- | --- | --- |
| Prior emissions (North America) | **2017** EPA platform (v7.1) | (Adams, 2020) |
| Prior emissions (China) | **2015** Tsinghua University | (Zhao et al., 2018) |
| Prior emissions (Rest of hemisphere) | HTAPv2 (2010) projected to **2014** using CEDS scaling factors | (Janssens-Maenhout et al., 2015; Hoesly et al., 2018) |
| Prior emissions (LNO$_x$) | **2017** GEIA* | (Price et al., 1997) |
| Biomass burning emissions | **2019** FINN* | (Wiedinmyer et al., 2011) |
| Soil NO$_x$ emissions | **2018** CAMS* v2.1 with canopy reduction factor | (Granier et al., 2019) |
| Biogenic emissions | **2018** MEGAN* | (Guenther et al., 2006) |
| Meteorology | **2019** WRF v4.1.1 | (Powers et al., 2017) |
| Satellite observation year | **2019** NO$_2$ retrievals from OMI and TROPOMI | |

* GEIA = Global Emissions Initiative; FINN = Fire Inventory from NCAR; CAMS = Community Atmosphere Modeling System; MEGAN = Model of Emissions of Gases and Aerosols from Nature



## 2.3 Chemical data assimilation in CMAQ

We adjust modeled $NO_2$ concentrations using satellite observations by coupling the CMAQ model to a data assimilation model, the National Centers for Environmental Prediction (NCEP) Gridpoint Statistical Interpolation (GSI) program version 3.3 (Shao et al., 2016). GSI performs three-dimensional variational (3DVAR) data assimilation by minimizing the cost
function, $J$:

$$J = \frac{1}{2}[x^T\mathbf{B}^{-1}x + (H(x) - y)^T\mathbf{R}^{-1}(H(x) - y)] , \qquad (1)$$

where $y$ is the observation innovation $y = y_o - H(x_b)$, $x$ is the analysis increment $x = x_a - x_b$, $x_a$ is the analysis field ($NO_2$ concentration after application of chemical data assimilation), $x_b$ is the model background (the simulated $NO_2$ concentration
before application of chemical data assimilation), $y_o$ is the satellite observations, $\mathbf{B}$ is the background error covariance matrix, $\mathbf{R}$ is the observation error matrix, and $H$ is the observation operator. To compute the difference between the model column ($x_b$) and the satellite column ($y_o$), the observation operator $H$ is applied, which transforms the model background to the form of the satellite observations. For TROPOMI data, the averaging kernel is first converted to scattering weights as

$$w(z) = \mathbf{A}(z) \times \mathbf{M}_{total} , \qquad (2)$$


where $\mathbf{A}(z)$ is the vertically-resolved TROPOMI averaging kernel for level $z$, $\mathbf{M}_{total}$ is the air mass factor provided with the satellite data, and $w(z)$ are the vertically-resolved scattering weights. Scattering weights accompany the OMI $NO_2$ data product, so this step is not needed to assimilate OMI data. Scattering weights are then applied to compute the model slant column as


$$\Omega_s^m = \sum_z \Omega_v^m(z)w(z), \ z \leq z_{tropopause}, \qquad (3)$$

where $\Omega_v^m(z)$ is the model partial vertical column in the troposphere, interpolated to the satellite grid, and $\Omega_s^m$ is the model tropospheric slant column density (SCD). The difference between the modeled and observed slant columns, or the observation innovation $y$ in Equation 1, is estimated as


$$\Omega_s' = \Omega_v^o\mathbf{M}_{trop} - \Omega_s^m, \qquad (4)$$

where $\Omega_s'$ is the analysis increment, $\Omega_v^o$ is the satellite tropospheric VCD, and $\mathbf{M}_{trop}$ is the tropospheric air mass factor, distributed with the satellite data. We eliminate the influence of the *a priori* satellite vertical profile by computing the



analysis increment with the model and observed SCD, which, unlike the VCD, does not rely on the *a priori* vertical $NO_2$

profile assumed by the satellite.

We compute **B** using the Generalized background error covariance matrix model (GENBE v2.0) (Descombes et al., 2015), which models background errors by comparing a free-running simulation and a simulation with either lightning or anthropogenic $NO_x$ emissions perturbed. We use GENBE with the prior simulation and a simulation with a uniform -15% perturbation to $LNO_x$ to create 3-dimensional background errors in the upper troposphere for the $LNO_x$ assimilation. After

updating $LNO_x$ emissions (as described in Sect. 2.5), we create 3-dimensional background errors in the boundary layer for the anthropogenic $NO_x$ assimilation by using GENBE with the $LNO_x$ posterior simulation and a simulation with a -15% perturbation to surface anthropogenic $NO_x$ emissions. Observation error **R** is provided with the satellite data.

Online coupling between GSI and CMAQ was developed in this study to perform the assimilation. At each model timestep in which a satellite observation is available, the CMAQ model simulation is paused and 3DVAR assimilation is

performed. The CMAQ model state at that time step is used as $x_b$. After assimilation using 3DVAR within GSI, CMAQ returns to a free-running mode and the new model state, $x$, is updated to more closely match the satellite observation. The difference in the monthly average $NO_2$ VCDs from the assimilation and no-assimilation runs is used in the inversion as $\Delta\Omega$.

### 2.4 Finite difference mass balance inversion

In the inversion framework developed, we iterate the approach of Lamsal et al. (2011). The FDMB process as applied here is summarized in Fig. 3. In the past, this approach has been used by directly comparing model and satellite columns (e.g. Itahashi et al., 2019; Cooper et al., 2017; Lamsal et al., 2011). We update the approach by first assimilating the satellite observations, and then updating the emissions using $\Delta\Omega$. All updates are performed on a monthly average basis.


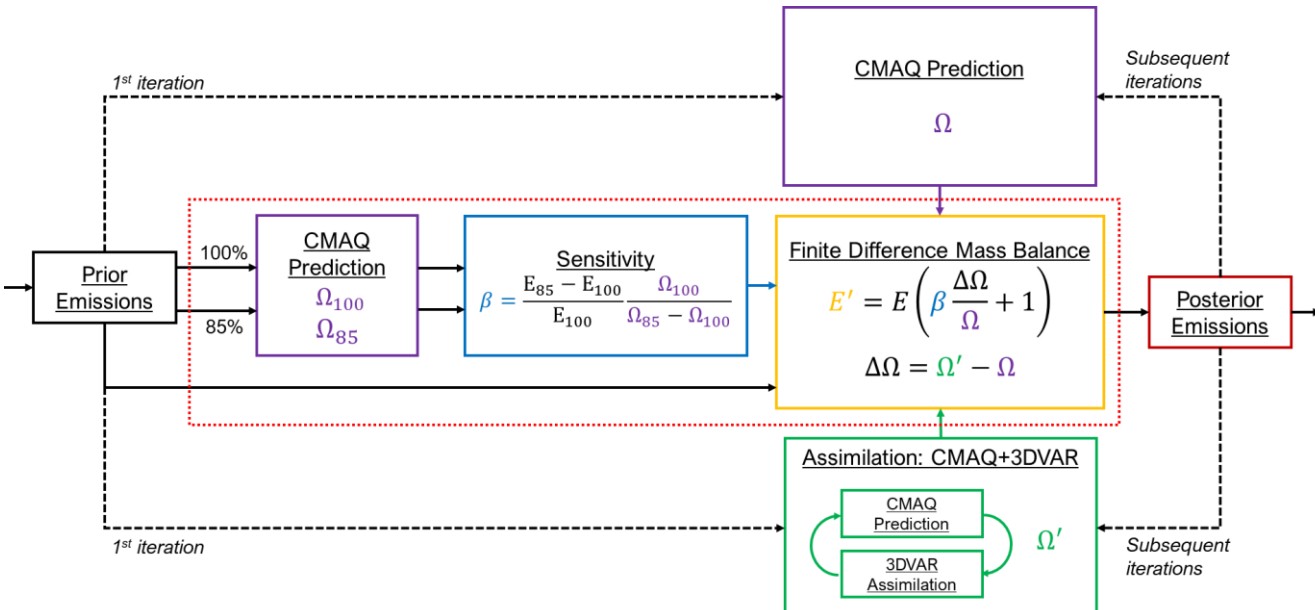

**Figure 3:** FDMB inversion. The red dashed line corresponds to the red dashed lines in Fig. 1, and the processes inside show additional details of the FDMB inversion. In this framework, the prior emissions (black box on the far left) are input to the CMAQ model. CMAQ simulations are performed with unperturbed prior emissions (100% arrow and $E_{100}$) and prior emissions with a -15% perturbation (85% arrow and $E_{85}$). The resulting modeled VCDs are $\Omega_{100}$ and $\Omega_{85}$, respectively. These VCDs are used to compute the sensitivity, $\beta$ (blue box). New emissions totals are calculated with FDMB (yellow box), using $\beta$, NO$_2$ VCD from a CMAQ simulation without assimilation ($\Omega$), and NO$_2$ VCD from a CMAQ simulation with assimilation ($\Omega'$). When iteration is used, the posterior emissions from the previous iteration are used as input to the CMAQ model to simulate new VCDs, $\Omega$ and $\Omega'$.

In FDMB, following Lamsal et al. (2011), emissions changes are inferred through the relationship

$$\frac{\Delta E}{E} = \beta \frac{\Delta \Omega}{\Omega} \qquad (5)$$

where $\Delta E$ is the inferred NO$_x$ emissions change, $E$ is the NO$_x$ emissions prior, $\Omega$ is the model simulated NO$_2$ VCD without chemical data assimiliation, and $\Delta \Omega = \Omega_{assim} - \Omega$, the monthly average difference between the model simulated tropospheric NO$_2$ VCD with ($\Omega_{assim}$) and without ($\Omega$) chemical data assimilation. $\beta$ is a unitless scaling parameter, the Jacobian, that linearly relates NO$_2$ VCD changes to NO$_x$ emissions changes. $\beta$ is calculated through finite differencing as

$$\beta = \frac{E'-E}{E} \frac{\Omega_E}{\Omega_{E'}-\Omega_E} \qquad (6)$$

where $E'$ is perturbed NOx emissions, $\Omega_E$ is the tropospheric NO$_2$ VCD simulated with model emissions $E$, and $\Omega_{E'}$ is the tropospheric NO$_2$ VCD simulated with model emissions $E'$. To estimate $\beta$, we use the same -15% perturbation used to create





background errors **B** in the boundary layer. Cooper et al. (2017) found that using perturbations ranging from 5% to 20% to calculate $\beta$ changed posterior emissions estimates by less than 2% globally.

### 2.5 Inverse modeling NOₓ emissions

In our framework, LNOₓ emissions are updated first. Due to the satellite instruments' sensitivity to NO₂ in the upper
atmosphere (e.g. Eskes and Boersma, 2003), small model biases there can influence the total column comparison and adversely impact the anthropogenic emissions adjustment. By updating LNOₓ emissions, we aim to decrease this bias and its impact on the ANOₓ inversion. We compute the scaling parameter for lightning emissions, $\beta_{LNOx}$, using the -15% LNOₓ perturbation simulations applied to create background errors for the upper troposphere. We then assimilate satellite NO₂ observations using the background errors for the upper troposphere and apply $\beta_{LNOx}$ in a single inversion iteration to
compute spatially-varying LNOₓ adjustment factors. Updates to LNOₓ are calculated using monthly averages.

ANOₓ emissions are updated by iteratively applying a FDMB inversion independently for each month in 2019. Iterating the FDMB has been shown to improve emissions estimates compared to a single FDMB application (Cooper et al., 2017). In the FDMB iteration, each update to the emissions serves as the prior emissions for the subsequent iteration (represented as black dashed lines in Fig. 1). The number of iterations is determined based on the synthetic observation experiment
described in Sect. 2.6. $\beta$ is held constant during all ANOₓ inversion iterations, and not recalculated each time, to prevent instability in $\beta$ as changes in the column become smaller with subsequent iterations. In the ANOₓ emissions inversion, we only consider grid cells in which local anthropogenic NOₓ emissions likely significantly contribute to the satellite-observed NO₂ column by only including grid cells in which anthropogenic NOₓ emissions comprise at least 50% of total NOₓ emissions following Lamsal et al. (2011), population density is greater than 15,000 people km⁻² (CIESIN, 2018), modeled
cloud cover is less than 30%, and the local time is 1:00 or 2:00 pm (OMI and TROPOMI overpass times). Only emissions in grid cells meeting these criteria are adjusted. Table S1 describes each simulation performed for the LNOₓ and ANOₓ inversions.

The FDMB method assumes emissions impacts are local (i.e., emissions in one grid cell do not affect VCD amounts in neighboring grid cells). This assumption is most valid when NOₓ lifetime to chemical losses is shorter than NOₓ transport
time to neighboring grid cells, which is typical near the surface in coarse resolution models (Martin et al., 2003), such as the one used in this study. However, the assumption is less realistic at finer resolutions and in the upper troposphere, where the lifetime of NO₂ is longer than at the surface and NO₂ concentrations are not directly impacted by coincident near-surface emissions. Even at coarse resolution (e.g., 100-km grid spacing), emissions smearing effects, which occur when the FDMB assumption of local emissions effects is incorrect and emissions are inappropriately adjusted, can appear due to NOₓ
transport, reservoir species, and chemical feedbacks (Turner et al., 2012; Cooper et al., 2017). Traditional FDMB, which directly compares modeled and remotely sensed columns, cannot address this effect. Assimilating the satellite VCD introduces an additional complication. The horizontal length scales (on the order of several hundreds of kilometers) used in





the background error extend beyond the grid cell horizontal dimensions (nominally 108km) in the middle and upper troposphere and, as a result, $NO_2$ changes introduced by assimilation ($\Delta\Omega$) do not have a local relationship with surface

emissions directly below. In our work, assimilating the observed column information, instead of directly comparing modeled and satellite retrieved VCDs, allows the analysis to be restricted to the lower troposphere, mitigating both the misallocation errors of FDMB and the effect of horizontal length scales extending beyond the grid cell dimension. To that end, we limit the anthropogenic emissions analysis to the lowest 20 model layers, which is nominally from the ground to ~720 hPa over non-mountainous terrain in the summer, and use that partial column to calculate $\Delta\Omega$ in the FDMB inversion. $\Delta\Omega$ for a single

month above and below the threshold is illustrated in Fig. S1. By applying this cutoff, we focus the inversion on surface anthropogenic $NO_x$.

**2.6 Inversion system testing**

We conduct a synthetic observation experiment to evaluate the ability of the inversion system to constrain emissions to a known perturbation. Artificial $NO_2$ observations were generated from CMAQ simulations with unperturbed emissions and

$NO_x$ emissions reduced by 15%. As expected, assimilating the synthetic observations derived from a simulation with unperturbed emissions results in an analysis increment of zero. The results of an iterative emissions inversion based on the synthetic observations derived from the simulation with perturbed emissions are shown in Fig. S2. Across Northern Hemisphere regions, the normalized mean error (NME) relative to the known perturbed emissions, and the rate at which it changes, decrease with subsequent iterations. The NME is minimized after 7–9 iterations, depending on the region. In all

subsequent results, emissions inferences made with 8 iterations of the inversion system are shown and analyzed. Convergence of the inversion in different global regions adds confidence to the system's ability to constrain real-world emissions.

**3. Results**

**3.1 Lightning $NO_x$ emissions updates**

Assimilation of retrievals from either satellite increases $LNO_x$ emissions across all seasons, relative to the prior emissions (monthly climatology from GEIA), with largest changes occurring during the summer (Figs. S3 and S4). Applying 2019 OMI data increases total $LNO_x$ emissions in 2019 by 20% over the GEIA climatology, while assimilation of TROPOMI data increases $LNO_x$ emissions by 24%. The emissions increases inferred by both satellite products are driven by $NO_2$ increases in the mid and upper troposphere due to assimilation, with changes near the surface negligible in comparison.

Increases in background areas with small $NO_2$ column totals and subsequent $LNO_x$ increases in these areas suggest a low bias in modeled background $NO_2$ relative to observations from both satellites. A low bias agrees with the findings reported by other model-satellite $NO_2$ comparisons (Silvern et al., 2019; Qu et al., 2021; Goldberg et al., 2017). The $LNO_x$ emissions





adjustments inferred here decrease the differences between modeled and satellite-derived NO$_2$ in the upper troposphere and decrease the bias that differences in the upper troposphere can introduce to the subsequent ANO$_x$ inversion.

### 3.2 Impact of assimilation on modeled NO$_2$ vertical column density

Figure 4 shows the change to CMAQ-modeled tropospheric VCD ($\Delta\Omega$) caused by assimilating NO$_2$ observations from OMI or TROPOMI with background errors for the boundary layer, before applying any emissions adjustments. In Fig. 4, and throughout the results, $\Delta\Omega$ reflects differences near the surface (as described in Sect. 2.5). Assimilating OMI NO$_2$ data generally increases modeled NO$_2$ columns near populated areas in China, India, and the U.S. In contrast, assimilating TROPOMI NO$_2$ data decreases modeled NO$_2$ columns more widely across the Northern Hemisphere. The changes brought about by assimilating satellite data are larger during the winter and fall, and smaller in the spring and summer, when NO$_x$ lifetime is shortest and NO$_2$ columns are smaller. During the winter in northeast China, where the assimilation impacts are most apparent, the seasonal average change due to assimilation reaches $1.8 \times 10^{15}$ molecules cm$^{-2}$ for OMI and $-2.8 \times 10^{15}$ molecules cm$^{-2}$ for TROPOMI. The direction of $\Delta\Omega$ after assimilation of OMI data is more heterogeneous and shows a stronger seasonality, while $\Delta\Omega$ based on assimilating TROPOMI data is consistently negative. Over Europe, $\Delta\Omega$ after assimilating OMI observations is close to zero in warm months and negative in colder seasons. Assimilating satellite-observed NO$_2$ increases the NO$_2$ levels modeled over the ocean and less-populous areas, such as the Sahara, with low NO$_x$ emissions and small NO$_2$ column amounts.



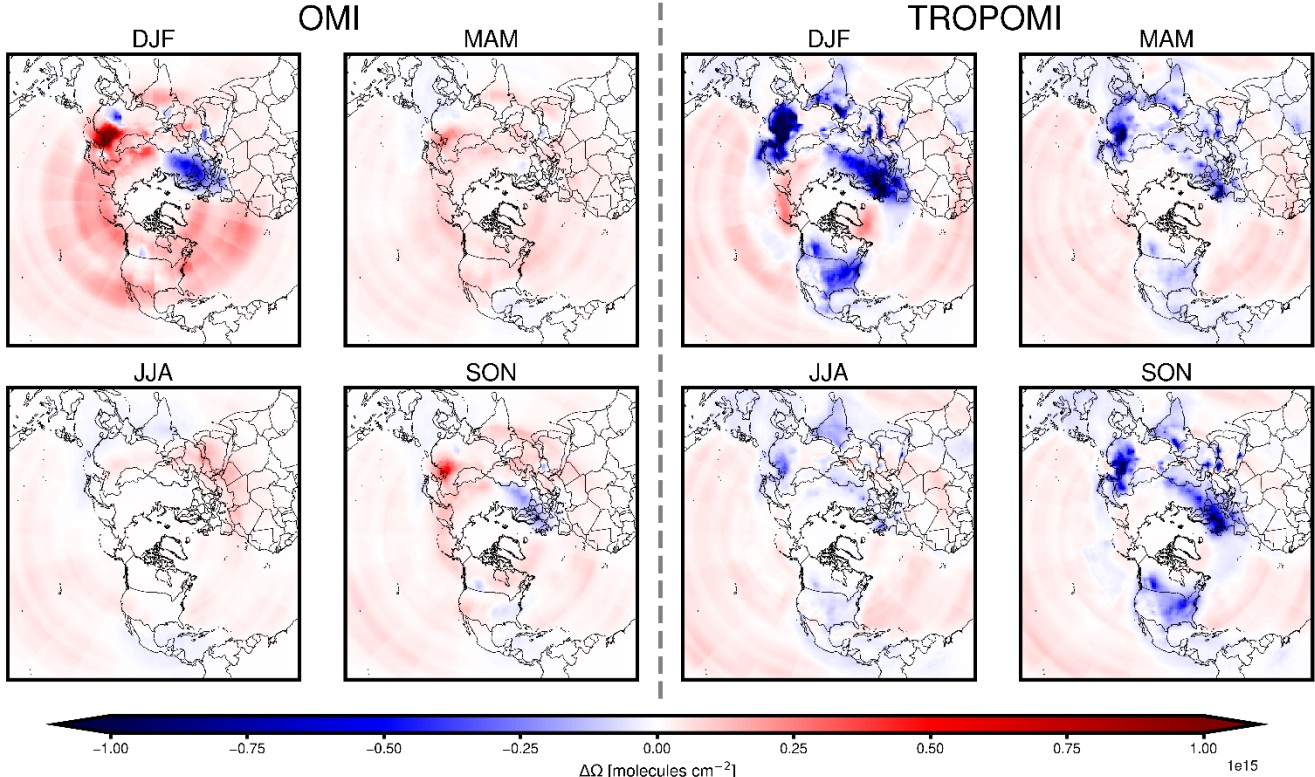


**Figure 4:** Seasonal NO$_2$ VCD change (**ΔΩ**) from CMAQ simulation using prior emissions after assimilating OMI or TROPOMI tropospheric NO$_2$ observations and modeling atmospheric composition with prior NO$_x$ emissions. **ΔΩ** shown for winter (DJF), spring (MAM), summer (JJA), and fall (SON).

Over polluted areas, the direction of ΔΩ for the TROPOMI or OMI data assimilations tends to differ. This discrepancy is likely due to the low bias in TROPOMI-derived tropospheric NO$_2$ columns, which has been reported to be approximately 10% over the U.S., Europe, and India, and greater than 20% over China when compared with the OMI Quality Assurance for Essential Climate Variables (QA4ECV) retrieval (Van Geffen et al., 2021; Verhoelst et al., 2021; Wang et al., 2020a; Li et al., 2021). Over background areas, the analysis increments that results from assimilation of observations from both satellites

generally agree. The consistency suggests a low bias in modeled background NO$_2$ concentrations and also agrees with the low bias in CMAQ-modeled free tropospheric NO$_2$ reported by Goldberg et al. (2017). Such a bias can contribute to the positive analysis increment over background areas. However, NO$_2$ columns observed in these regions may be smaller than the retrieval accuracy of $0.7 \times 10^{15}$ molecules cm$^{-2}$ (Van Geffen et al., 2019), reducing confidence in the analysis increment at these locations. In the anthropogenic emissions inversion, our filtering criteria exclude from the analysis background areas

which are more likely to have low VCD amounts.



### 3.3 Emissions inversion

Season-average $\beta$ values, relating $NO_2$ vertical column differences to anthropogenic near-surface $NO_x$ emissions updates, are shown in Fig. S5. Based on our criteria for gridcell inclusion in the inversion, described in Sect. 2.5, we consider
13% of the grid cells in the domain, which represent 88% of prior anthropogenic $NO_x$ emissions. Seasonal domain-average values range from 1.33 to 1.66, and are lower in the winter and higher in the summer. A $\beta$ value less than 1.0 results in an emissions update that is smaller than the VCD change, while a $\beta$ greater than 1.0 has the opposite effect. $\beta$ tends to be less than 1.0 in polluted regions during colder months and larger during warmer months and in less polluted regions, although many grid cells which are less polluted are not considered in the analysis. The scaling factors are smallest over China, and
larger over the U.S., India, Mexico, and Europe. The differences among regions stem from local differences in $NO_x$ lifetime and transport. In Indonesia and sub-Saharan Africa, lower emissions and a small response from tropospheric VCD to anthropogenic emissions perturbations can lead to large $\beta$ values. To prevent overly large or small $\beta$ values, we constrain the factor to between 0.1 and 10, following Cooper et al. (2017). Scaling factors estimated here are larger than the 1.16 global-average previously reported by Lamsal et al. (2011). However, in Lamsal et al. (2011) modeled $NO_2$ vertical columns were
sampled at the morning SCanning Imaging Absorption spectroMeter for Atmospheric CHartographY (SCIAMACHY) overpass time, rather than the afternoon OMI or TROPOMI overpass times, and $\beta$ tends to be closer to 1.0 during the morning in regions with high $NO_x$ emissions (Li and Wang, 2019). Li and Wang (2019) show that over rural regions with lower $NO_x$ concentrations, $\beta$ is larger at the OMI or TROPOMI overpass window than at the SCIAMACHY overpass window, suggesting a larger overall $\beta$ for analyses based on OMI or TROPOMI products should be expected. Additionally,
$NO_x$ emissions have decreased considerably in several regions of the Northern Hemisphere after the Lamsal study was conducted (2011), including the U.S. (Tong et al., 2015) and China (Miyazaki et al., 2017), which has changed the sensitivity of $NO_2$ VCDs to $NO_x$ emissions (Qu et al., 2021; Silvern et al., 2019).

Annual bottom-up prior $ANO_x$ emissions estimates are shown in Fig. 5. Season-average $ANO_x$ emissions inferences from the inversions based on OMI and TROPOMI observations are shown in Fig. 6. The use of OMI observations generally
tends to increase emissions in most industrialized nations outside of Europe. $NO_x$ emissions increases driven by OMI observations are largest in winter and spring and smaller, or slight decreases, in summer and fall. In contrast, the use of TROPOMI retrievals tends to drive a decrease in $NO_x$ emissions across all seasons and continents, with the largest impacts in the summer and smallest in the spring. The largest emissions changes based on both OMI and TROPOMI retrievals are in Northeast China during the winter. Over India, OMI inferred changes are concentrated in Central India where prior
emissions are lower, while the largest changes inferred from TROPOMI are in the Northern, Eastern, and Southern zones, where prior emissions are highest. Relative $NO_x$ emissions changes driven by TROPOMI observations tend to be small over dense urban areas, with more uniform decreases over cells with lower emissions.

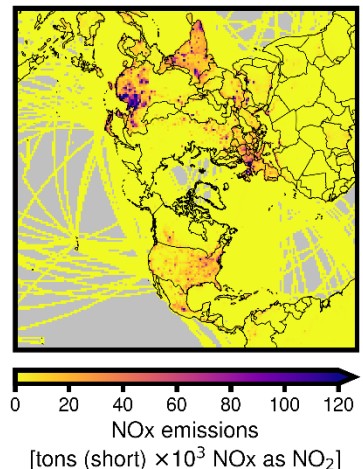

**Figure 5:** 2019 prior anthropogenic $NO_x$ emissions totals. Data sources are described in Table 1.


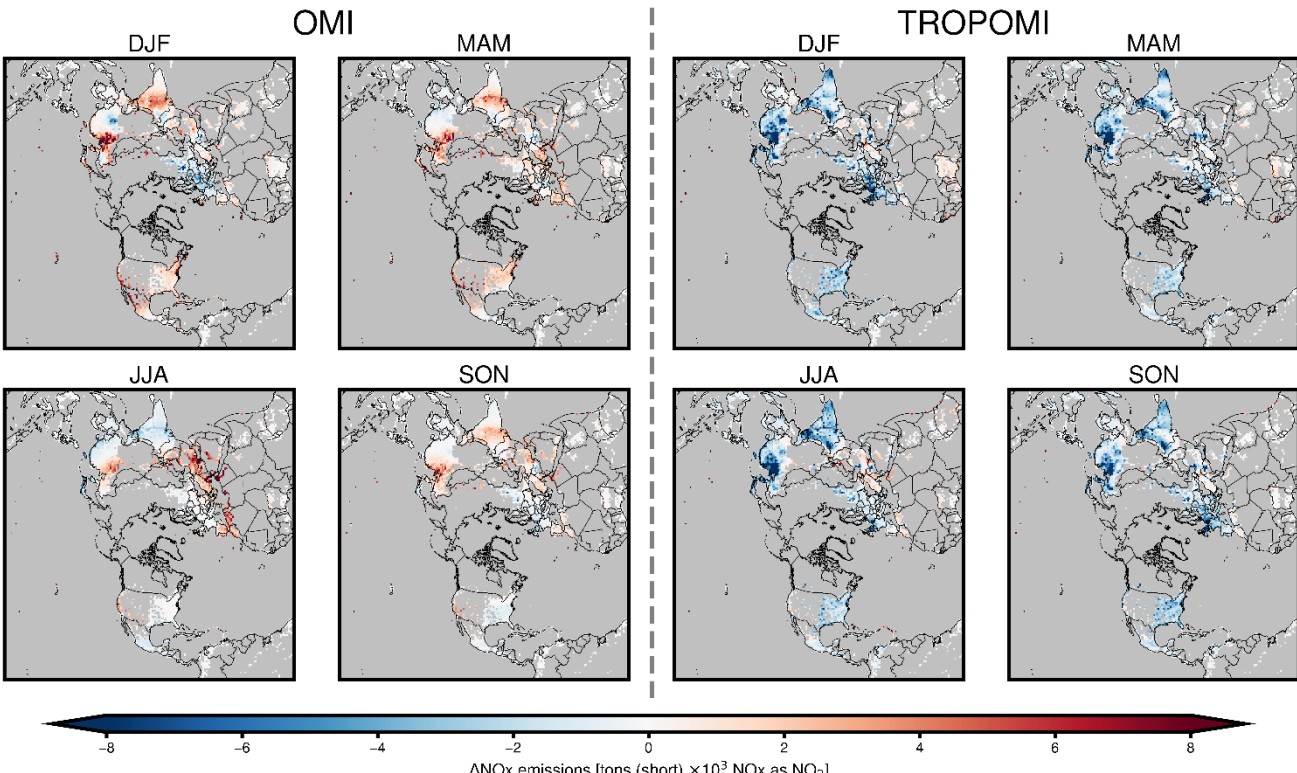

**Figure 6:** Season-average $NO_x$ emissions changes from inverse modeling updates based on OMI and TROPOMI observations. Emissions changes are shown for winter (DJF), spring (MAM), summer (JJA), and fall (SON).





ANOx emissions totals and inferred changes are explored for China, India, Europe, Mexico, and the U.S. (Fig. 7). We also show 2019 $NO_x$ emissions totals from the Copernicus Atmosphere Monitoring Service (CAMS) bottom-up emissions inventory (Granier et al., 2019), and the NASA Tropospheric Chemical Reanalysis products 2 (TCR-2) satellite-inferred inventory (Miyazaki et al., 2019, 2020). TCR-2 top-down NOx emissions are constrained using satellite observations of $NO_2$, CO, $O_3$, and $SO_2$ at a resolution of $1.125° \times 1.125°$ and are further described in (Miyazaki et al., 2017). CAMS

anthropogenic $NO_x$ emissions are based on the Emissions Database for Global Atmospheric Research (EDGAR version 5.3) estimates for 2015 (Crippa et al., 2020) projected to 2019 using CEDS scaling factors, and are provided at $0.1° \times 0.1°$. Both datasets provide monthly anthropogenic $NO_x$ totals. Except for Europe, assimilation toward OMI retrievals increases annual emissions totals in the regions analyzed, while using TROPOMI retrievals decreases them. TCR-2 $NO_x$ emission estimates are larger than the prior emissions used by our inverse modeling framework, except for India, while CAMS totals are lower

than the prior emission estimates and similar to TROPOMI inferred emissions. Across the regions considered, TROPOMI infers an average annual decrease to $NO_x$ emissions from the regions of -33%, while OMI infers a +9% increase. In Europe, the only region where the sign of the inferred changes match, use of OMI retrievals results in a -1% change, while applying TROPOMI observations leads to a -36% decrease in $NO_x$ emissions. The largest total changes are inferred in the highest emitting region, China, while the greatest relative changes, -41% inferred with TROPOMI, are for India, where emissions

are highly uncertain. Changes inferred with OMI observations over the U.S. are greater than $1,200 \times 10^3$ short tons $NO_x$ as $NO_2$ per year, but smaller than the difference between our prior U.S. emissions estimates and TCR-2 or CAMS estimates. A change of $-3,000 \times 10^3$ short tons $NO_x$ as $NO_2$ emitted annually in the U.S. as inferred by TROPOMI, over 30% of the prior emissions, differs significantly from National Emissions Inventory estimates but leads to a total close to that of the 2019 CAMS inventory.

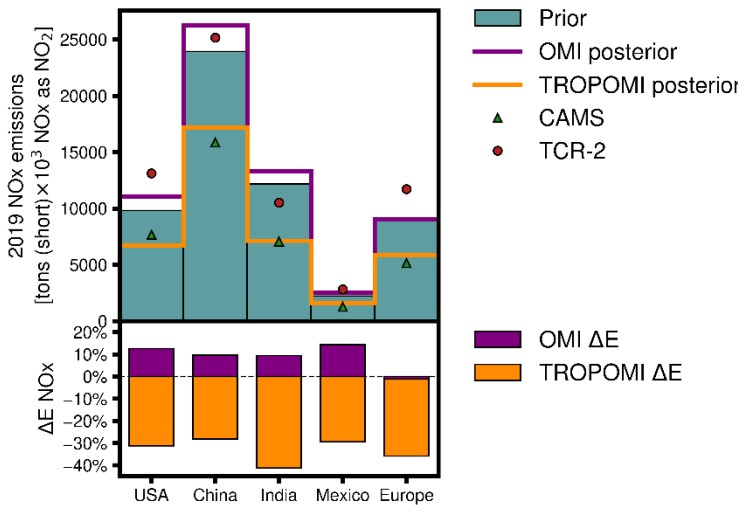


**Figure 7:** Prior and satellite-inferred 2019 anthropogenic NOx emissions in select global regions. Top plot shows total emissions (as NO₂) from prior emissions estimates, inference with OMI or TROPOMI observations (OMI and TROPOMI posterior), and CAMS or TCR-2





inventories in the U.S., China, India, Mexico, and Europe. The bottom plot shows the percent change ($\Delta E$ NO$_x$) inferred with OMI or TROPOMI data, relative to prior emission estimates, for each region.


Across the months simulated, inferences using OMI retrievals consistently lead to higher NO$_x$ emissions than using TROPOMI retrievals. Figure 8 shows monthly NO$_x$ emission totals and inferred changes for several global regions. The magnitude of changes is generally smallest in summer months and largest in winter months for both OMI and TROPOMI inferred emissions. Monthly prior emissions totals lay between the OMI and TROPOMI inferences, except for summertime

emissions in India and Mexico where both satellite inferences decrease NO$_x$ emissions. Over Europe, both satellite products infer a decrease during the winter, although fewer valid satellite pixels due to snow cover at high latitudes and longer winter NO$_2$ atmospheric lifetimes may influence the inference.

**Figure 8:** Monthly prior and satellite-inferred anthropogenic NO$_x$ emissions in 2019 in select global regions. Total monthly emission (as
NO$_2$) from prior emission estimates, inference with OMI or TROPOMI observations (OMI and TROPOMI posterior), and CAMS or TCR-





2 inventories in the U.S., China, India, Mexico, and Europe are shown. Percent change (ΔE NOx) inferred with OMI or TROPOMI data, relative to prior emission estimates, for each region are shown by the purple and orange bars.

Based on reported $NO_x$ emissions trends (McDuffie et al., 2020; U.S. EPA, 2022a), changes from the prior emissions

inventory (Table 1) to 2019 are expected. Relative to the prior emissions, significant decreases in $NO_x$ emissions in China, estimated for 2015 in the prior inventory, and smaller reductions in Europe and North America, reported for 2014 and 2017 in the prior inventory, respectively, should be anticipated. TROPOMI inferred emissions reflect the direction anticipated for these changes, but with larger than expected magnitudes. For example, the 28% decrease in anthropogenic $NO_x$ emissions over China between 2015 and 2019 inferred from the TROPOMI observations is substantially larger than the 8% decrease

estimated between 2015 and 2017 by the Community Emissions Data System (McDuffie et al., 2020). Bottom-up estimates indicate that anthropogenic $NO_x$ emissions in the U.S. have decreased through 2019 (U.S. EPA, 2022a). Although the direction of the emissions change inferred from TROPOMI agrees with the trend in bottom-up estimates, its magnitude is larger than expected. An underestimate of U.S. emissions in winter in the prior inventory when compared with OMI inferences contrasts with field study results reporting no bias in Northeastern U.S. winter emissions estimates (Jaegle et al.,

2018; Salmon et al., 2018). In India, bottom-up emissions inventories report sustained growth of $NO_x$ emissions (Kurokawa and Ohara, 2020; McDuffie et al., 2020) and $NO_2$ levels observed by OMI have been increasing since 2005 (Goldberg et al., 2021; Cooper et al., 2022). The decrease in anthropogenic $NO_x$ emissions inferred by TROPOMI observations contrasts with these trends in bottom-up estimates and OMI observations.

The low bias known to affect TROPOMI $NO_2$ observations influences the results of the emissions inversion, which

targets grid cells with high emissions, likely leading to larger than expected inferred emissions decreases. We conduct an inversion using the reprocessed TROPOMI $NO_2$ version 2.3.1 (Van Geffen et al., 2021) to infer $NO_x$ emissions for January 2019, and find that the updated data increases the TROPOMI posterior inference by 17% over the U.S. and 4% in China relative to version 1.2.2, but still differs from significantly from that obtained using OMI observations (Figs. S10 and S11). The differences between emissions inferred by OMI and TROPOMI observations highlight the importance of ongoing

efforts to harmonize OMI and TROPOMI $NO_2$ retrieval algorithms, such as the NASA Multi-Decadal Nitrogen Dioxide and Derived Products from Satellites (MINDS) (Lamsal et al., 2020) and the QA4ECV (Boersma et al., 2017) datasets.

In addition to smearing effects, coarse resolution models can artificially alter nonlinear $NO_2$ chemistry, leading to biases in inferences of $NO_x$ emissions from satellite $NO_2$ columns (Valin et al., 2011; Sekiya et al., 2021; Lamsal et al., 2011). Higher resolution simulations can better resolve $\beta$ and reduce biases caused by nonlinear chemistry. Additional errors in the

emissions estimates may be associated with emissions from non-anthropogenic $NO_x$ sources. Although the emissions inversion targets anthropogenic sources only, changes in $NO_2$ columns observed by the satellite instruments driven by natural $NO_x$ emissions processes may not be captured by in the air quality model simulations and subsequently lead to biased anthropogenic emissions inferences (Li et al., 2021).



The emissions resulting from the inverse modeling framework are comparable to CAMS and TCR-2 2019 emissions
estimates in several ways. In the U.S., China, and Europe, the magnitudes of $ANO_x$ emissions from OMI retrievals are
comparable to TCR-2 $NO_x$ emissions estimates and exhibit similar monthly patterns. Annual $NO_x$ emissions inferred from
OMI observations are also relatively similar to TCR-2 estimates for India and Mexico, although monthly emissions patterns
differ. Unlike TCR-2 emissions estimates, which are is also constrained by OMI $NO_2$ observations, 2019 CAMS emissions
estimates are projected from 2015 bottom-up data. However, CAMS estimates provide a representation of anticipated
emissions trends. In all regions considered, CAMS $NO_x$ emissions estimates are close to the TROPOMI inference annual
totals and lower than the prior emissions, OMI inferences, and TCR-2 estimates, potentially suggesting that global $NO_x$
emissions have not decreased as much as anticipated by the CAMS inventory projections.

**3.4 Impacts of emissions updates on modeled $NO_2$ and $O_3$**

We evaluate and compare the CMAQ simulations' ability to reproduce observed pollutant concentrations when driven
with $NO_x$ emissions estimates from the prior inventory and those derived by the inverse modeling framework. Figure 9
compares 2019 OMI and TROPOMI $NO_2$ VCD retrievals with modeled $NO_2$ VCDs using the prior emissions with no
updates, $LNO_x$ emissions updates, and $LNO_x$ and $ANO_x$ emissions updates. Satellite-based $LNO_x$ emissions updates improve
CMAQ model performance—correlation coefficient (R), normalized mean error (NME), and normalized mean bias
(NMB)—when evaluated against tropospheric VCD retrievals, relative to model performance with the prior emissions. OMI-
inferred $ANO_x$ emissions updates further improve CMAQ model performance evaluated against VCD retrievals, decreasing
NMB from -20% to -5% and NME from 38% to 28%. Model performance is improved by using OMI data in the inverse
modeling framework across all seasons (Figs. S6-S9). Although $LNO_x$ emissions updates derived from TROPOMI
observations improve model bias and error relative to the CMAQ simulation using prior emission estimates, TROPOMI-
inferred anthropogenic emissions do not, except during summer months (Figs. 9 and S6-S9). The lack of significant
improvements in CMAQ-simulated $NO_2$ VCDs after applying the emissions inversion with TROPOMI $NO_2$ retrievals prior
to the version 2.3.1 update (Van Geffen et al., 2021) may be associated with changing chemical regimes that are not captured
in the emissions inversion process.

    Changes in modeled VCD due to assimilation and the emission inferences calculated in the TROPOMI $ANO_x$ inversion
exceed the emissions perturbation and VCD changes used to calculate $\beta$. For example, over the eastern U.S., the -15%
emissions perturbation used to calculate $\beta$ leads to VCD changes of -15% on average in winter, but assimilating TROPOMI
retrievals leads to VCD changes ($\Delta\Omega$) of -19% on average in the winter, with individual changes exceeding -30%. Modeled
$NO_x$ chemistry and $NO_2$ vertical profiles after assimilating TROPOMI retrievals may be different than those used in the
calculation of $\beta$. As a result, assimilating TROPOMI retrievals in the ANOx inversion may lead to modeled $NO_2$ vertical
profiles which are inconsistent with the precalculated $\beta$ used in the FDMB relationship and less reliable subsequent
emissions inferences. In contrast, the magnitude of VCD changes due to assimilating OMI retrievals over the eastern U.S. in





winter is 8%, well within the magnitude of the VCD changes used to precalculate $\beta$. This highlights the importance of applying a $\beta$ sensitivity valid for the magnitude of anticipated emissions changes in FDMB inversions and the potential consequences of relying on satellite-derived retrievals with pre-existing biases in emissions inversions.


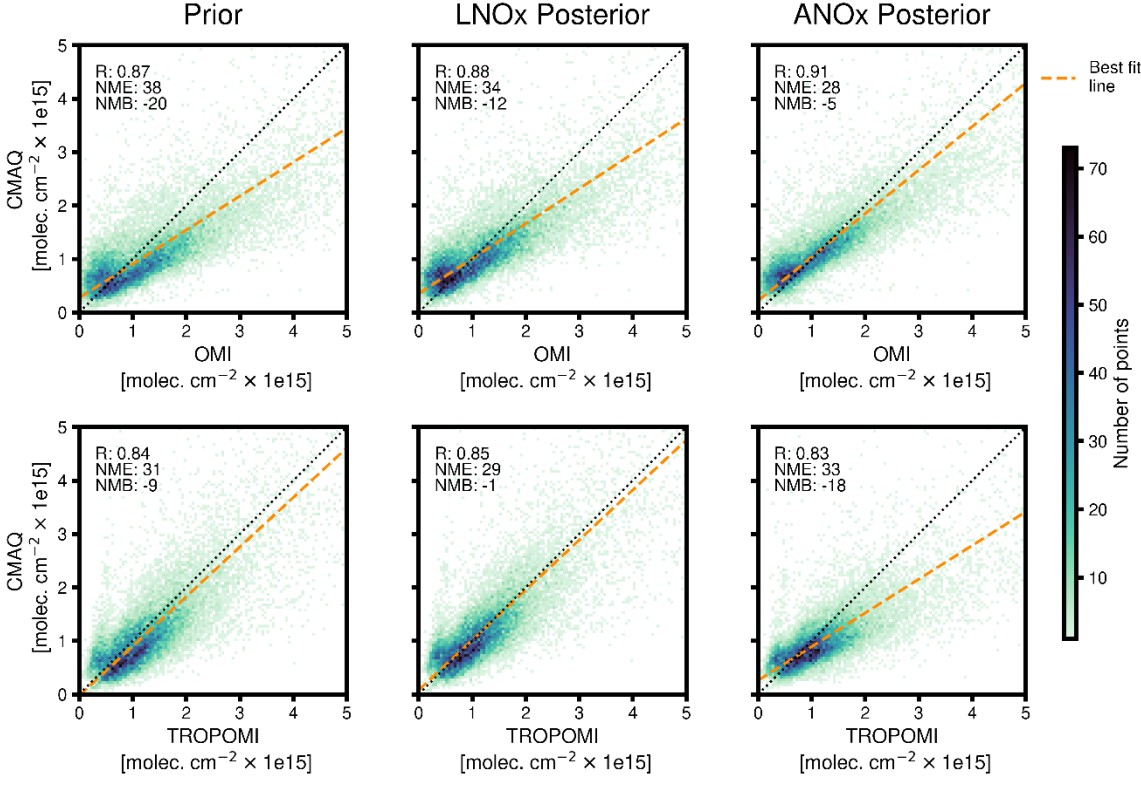

**Figure 9:** Impact of NOx emissions updates on modeled NO₂ VCDs. Plots compare 2019 season-average CMAQ-modeled NO₂ VCD at each model grid cell in which NOx emissions were updated by the inverse modeling framework against OMI and TROPOMI tropospheric NO₂ VCD retrievals averaged in each model grid cell. Modeled NO₂ VCD using prior emissions (Prior), inferred LNOx emissions (LNOx posterior), and inferred lightning and anthropogenic NOx emissions (ANOx posterior) are each compared with NO₂ VCD retrievals. Top row plots compare retrievals and modeled VCD based on OMI observations, while bottom row plots compare retrievals and modeled VCD based on TROPOMI observations. Linear regression line, correlation coefficient (R), normalized mean error (NME), and normalized mean bias (NMB), relative to tropospheric NO₂ VCD retrievals, are shown for each CMAQ simulation.

Comparing CMAQ-modeled $O_3$ to ozonesonde measurements from the World Ozone and Ultraviolet Radiation Data Centre (WOUDC) network shows the impacts updating $LNO_x$ emissions on simulated tropospheric $O_3$ (Fig. 10). Above 300 hPa, the model is biased low, but neither update has a major impact on this bias. However, within the free troposphere, the effects of $LNO_x$ emissions updates are larger. $LNO_x$ satellite-inferred emissions from both satellites increase $O_3$ and subsequently improve the model's low $O_3$ bias across all seasons, with the strongest effect in the summer. This suggests a



low background $NO_2$ in our prior simulation, consistent with several studies demonstrating that models underestimate background $NO_2$ (Goldberg et al., 2017; Qu et al., 2021; Silvern et al., 2019).

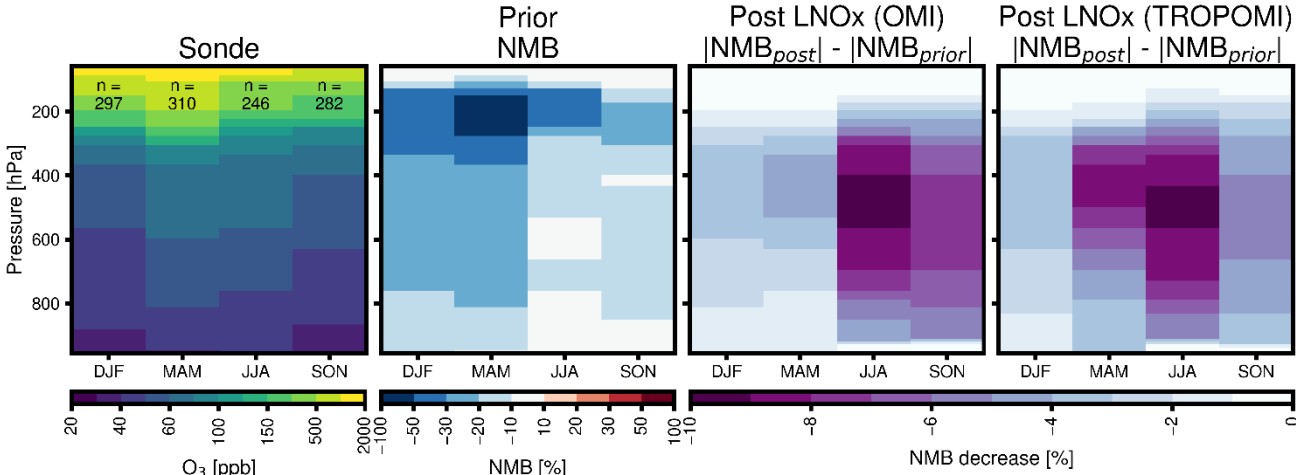

**Figure 10:** Ozonesonde observations from the WOUDC network and impact of lightning emissions inferences on modeled ozone. Left plot shows sonde observations averaged in each season and total number of launches per season. The NMB is shown for the prior emissions simulation. Plots on the right show the decrease in the NMB, relative to the prior simulation, for simulations with $LNO_x$ emissions updated with OMI and TROPOMI data.

Comparisons of CMAQ-modeled $NO_2$ and $O_3$ concentrations with ground-level measurements highlight the challenges of reproducing local air quality with a coarse scale model, but suggest potential to improve model performance with satellite-derived $NO_x$ emissions updates. Table 2 shows statistics evaluating modeled ground-level daily average $NO_2$ and maximum 8-hour $O_3$ concentrations over the U.S. against observations from 1,218 monitoring sites in the Air Quality System (AQS) (U.S. EPA, 2022b), excluding near-road monitors for which the gridded $NO_2$ fields are not representative of. Statistics for each season are included in Tables S2 and S3. There is a significant low bias in CMAQ-predicted ground-level $NO_2$ concentrations compared with monitoring site measurements, likely due to the model's coarse grid resolution and the aggregation of $NO_2$ monitors within urban areas with high $NO_x$ emissions and large concentration gradients. CMAQ simulations at higher horizontal resolution do not show the same bias against $NO_2$ surface observations (Toro et al., 2021). Agreement between modeled and observed ground-level $NO_2$ concentrations is improved by using OMI-inferred $NO_x$ emissions, compared with the prior emissions simulation, in particular during winter and spring months. Model performance evaluated against ground-level $O_3$ measurements improves to a smaller extent with OMI-inferred $NO_x$ emissions during winter and spring months. The use of TROPOMI-inferred emissions has mixed impacts on CMAQ performance against observed ground-level $NO_2$ and $O_3$ concentrations, leading to limited gains in seasonal R and some seasonal biases and errors, but also less agreement with observations for other seasonal statistics. In the U.S., the network of ground-based air quality observations is relatively large. However, in some regions where emissions uncertainties are expected to be





especially high, ground-based observations are significantly limited and less accessible. Assessing the impact of emissions updated against ground-based observations in these regions, although a challenge, would provide further evaluation of the inversion framework in locations where satellite retrievals have the largest potential to provide important constraints to emissions estimates.

**Table 2:** CMAQ model performance evaluated against daily average $NO_2$ (DA $NO_2$) and maximum 8-hour $O_3$ concentrations (MDA8 $O_3$) observed in 2019 by AQS monitoring sites in the U.S. Near-road monitors are not considered. Statistics are shown for simulations using prior emissions (Prior), lightning and anthropogenic NOx emissions inferred with OMI data (OMI-inferred), and lightning and anthropogenic NOx emissions inferred with TROPOMI data (TROPOMI-inferred). Coefficient of determination (R), normalized mean error (NME), and normalized mean bias (NMB), relative to AQS observations, are estimated for each CMAQ simulation.

| Pollutant | $NO_x$ emissions | R | NME | NMB |
|---|---|---|---|---|
| MDA8 $O_3$ | Prior | 0.65 | 15.9% | -1.4% |
| | OMI-inferred | 0.68 | 15.4% | 3.4% |
| | TROPOMI-inferred | 0.68 | 15.3% | -3.3% |
| DA $NO_2$ | Prior | 0.45 | 62.2% | -56.9% |
| | OMI-inferred | 0.52 | 57.4% | -49.6% |
| | TROPOMI-inferred | 0.45 | 71.7% | -69.9% |

**3.5 Impacts of emissions updates on long-range $O_3$ transport**

Global $NO_x$ emissions estimates affect model simulations of long-range air pollution transport. To explore these impacts, we examine the response of CMAQ-modeled transpacific $O_3$ to the inverse modeling framework's $NO_x$ emissions updates. Figure 11 shows season-average changes in simulated free-tropospheric $O_3$ over the North Pacific Ocean resulting from the use of OMI- and TROPOMI-inferred $ANO_x$ emissions, relative to the emissions simulation with $LNO_x$ emissions

updated. As expected, the emissions inversions lead to $O_3$ variations that follows $NO_x$ emissions changes inferred for each satellite's observations, with OMI inferences resulting in higher $O_3$ concentrations and TROPOMI inferences resulting in lower $O_3$ concentrations over the North Pacific Ocean. Season-average differences with respect to the prior emissions simulation are as large as +1.8 ppb in winter, using OMI-based updates, and -1.9 ppb in spring, using TROPOMI-based updates. Combined with transpacific wind patterns, the effects of the $NO_x$ emissions inversions on modeled $O_3$ suggest

potential implications of uncertain Asian emissions estimates for U.S. air quality management and emphasize the impacts of biases in satellite retrievals in inverse modeling systems.

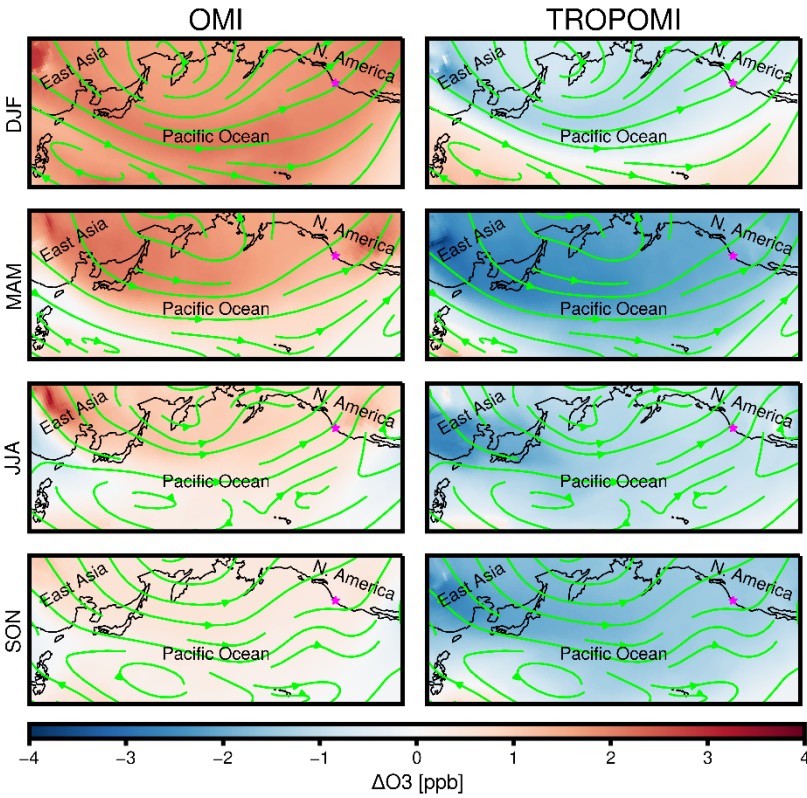

**Figure 11:** Changes 2019 season-average free-tropospheric $O_3$ concentrations (averaged between 750-250 hPa) simulated over the North Pacific Ocean using lightning and anthropogenic $NO_x$ emissions inferred with OMI or TROPOMI observations, relative to simulation using prior $ANO_x$ emissions and updated $LNO_x$ emissions. Differences are shown for winter (DJF), spring (MAM), summer (JJA), and fall (SON). Arrows depict season-average free-tropospheric winds (750-250 hPa). Star marker indicates location of Trinidad Head, California.

At the Trinidad Head, California, a location where atmospheric composition is relatively unaffected by local emissions sources and responsive to transpacific pollution transport (Fig. 11), differences in modeled daily average free-tropospheric $O_3$ concentrations can reach +5 ppb or -3 ppb. Figure 12 compares CMAQ-modeled vertical $O_3$ profiles to observations from 39 ozonesondes launched in at Trinidad Head in 2019 (WOUDC, 2019). Relative to the CMAQ simulation using prior emissions, $NO_x$ emissions updates inferred from OMI and TROPOMI data can improve the model's ability to reproduce ozonesonde $O_3$ distributions measured from the site, in particular during winter and spring when the discrepancies between modeled and observed concentrations are largest. These impacts on modeled vertical $O_3$ profiles are largely driven by changes the modeling framework's updates to lightning $NO_x$ emissions. The inferred $LNO_x$ increases from each satellite improve $O_3$ biases, while subsequent anthropogenic updates have smaller impacts, suggesting that biases in $O_3$ could be driven by background $NO_2$ composition in the model and not solely by long-range transport resulting from anthropogenic emissions.





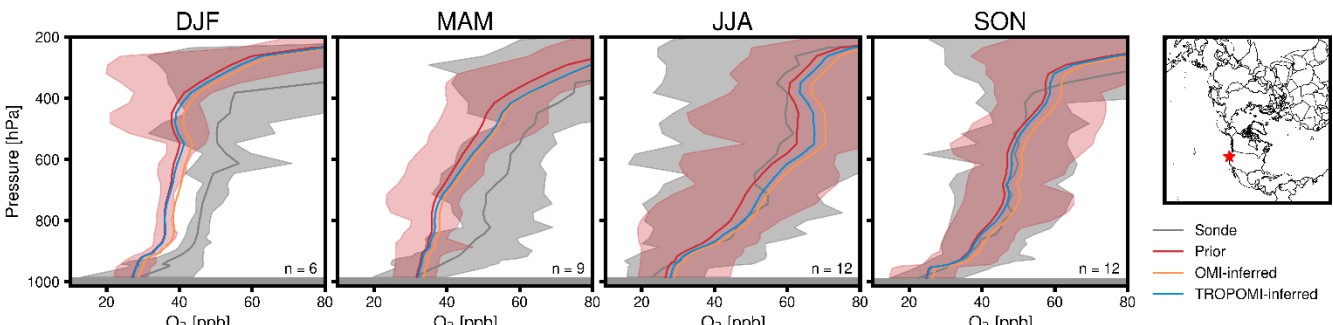

**Figure 12:** Season-average vertical O$_3$ concentration profiles modeled by CMAQ and measured by ozonesondes launched at Trinidad Head, California, in 2019. Vertical distributions are shown for simulations using prior emissions (Prior), lightning and anthropogenic NOx emissions inferred with OMI data (OMI-inferred), and lightning and anthropogenic NOx emissions inferred with TROPOMI data (TROPOMI-inferred). Modeled season-average profiles are shown during winter (DJF), spring (MAM), summer (JJA), and fall (SON) for days and times matching ozonesonde launches. Shading around sonde and prior emissions profiles represent the maximum and minimum O$_3$ at each pressure level. Map shows location of the Trinidad Head launch site.

## 4 Conclusions

In this study, we describe a satellite chemical data assimilation and inverse emissions modeling framework based on the CMAQ hemispheric air quality modeling platform. In the framework, data assimilation adjusts modeled NO$_2$ concentrations online using satellite retrievals of tropospheric NO$_2$ VCDs. The NO$_2$ column changes drive the FDMB inversion, resulting in satellite-constrained top-down emissions estimates. Here, we implement the framework in a NO$_x$ emissions inversion to update 2019 Northern Hemisphere NO$_x$ emissions estimates using NO$_2$ products from the OMI and TROPOMI satellite instruments. Relative to the modeling platform's prior emissions derived from regional and global emissions inventories, updates inferred using OMI and TROPOMI observations change average anthropogenic NO$_x$ emissions by –41% to +12% in China, the U.S., India, Europe, and Mexico. Evaluated against ground-based NO$_2$ observations recorded over the U.S. in 2019, the model performs best when using OMI-updated emissions, although a low bias in CMAQ predictions using prior emissions persists into simulations with satellite data assimilation. Compared with U.S ground-based O$_3$ observations, satellite-inferred emissions have mixed impacts on model performance, improving agreement with the measurements during certain months. LNO$_x$ emissions inferences improve modeled O$_3$ when compared against ozonesonde observations across the Northern Hemisphere. The framework's NO$_x$ emissions updates also affect model estimates of trans-Pacific O$_3$ transport, a source of growing concern in the U.S., with changes ranging from -3 ppb to +5 ppb in simulated O$_3$ at a remote West Coast site resulting from use of satellite-inferred emissions.

The modeling framework presented has several limitations. The global coverage of instruments on polar-orbiting satellites, such as Aura and Sentinel-5P, makes the emissions inversions possible but does not allow satellite observations to inform diurnal emissions variations. Upcoming geostationary satellite missions, including GEMS, TEMPO, and Sentinel-4, will provide this capability. Our approach, which balances computational costs and precision in the inversion, is subject to several assumptions. Large changes to the model concentrations resulting from the chemical data assimilation may invalidate



assumptions in the subsequent FDMB inversion, leading to biases in the inferred emissions. The FDMB inversion treats each grid cell independently and cannot relate $NO_2$ column changes in one grid cell to emissions in another. Although emissions smearing in the approach is mitigated by only analyzing the lower portion of the model column, our emissions changes may be less precise than targeted assimilation methods, such as 4DVAR adjoint-based methods. Further, coarse grid resolution

exacerbates biases in modeled $NO_2$ columns (Valin et al., 2011) and inferred $NO_x$ emissions (Sekiya et al., 2021). The air quality model used here does not include stratospheric chemistry, which could affect comparisons against $NO_2$ retrievals. Nevertheless, the framework shows the potential to improved air quality model predictions using satellite-derived emissions updates, in particular for regions with highly uncertain emissions inventories.

Emissions inversions based on satellite observations can provide valuable information for air quality modeling by

addressing the gaps in bottom-up emissions inventories. However, our analysis shows that such inversions and subsequent air quality simulations can be strongly influenced by uncertainties and biases in the satellite data products used. In the analysis conducted, $NO_x$ emissions inferred from TROPOMI observations appear biased low when assessed against those inferred from OMI data and surface and concentration measurements. The bias is consistent with recent research showing a low bias in TROPOMI v1.2 and v1.3 tropospheric columns (Judd et al., 2020; Verhoelst et al., 2021; Li et al., 2021; Van

Geffen et al., 2021). The results highlight the importance of efforts to develop robust and consistent satellite data products for use in air quality modeling evaluation, assimilation, and emissions inversions. Ongoing efforts to this end include the MINDS (Lamsal et al., 2020) and the QA4ECV (Boersma et al., 2017) projects. This study also emphasizes the need for longer-term satellite data assimilation and comparisons of established and new satellite data products. The framework introduced here can serve a generalized tool with applications beyond those explored in this study, and allows new satellite

data products to be incorporated as they become available. As satellite data products evolve and advance, the emissions inferred by the framework will improve.

*Data and code availability.* $NO_x$ emissions data derived from this research are available from the authors upon request. Level-2 satellite retrievals are available from NASA's Goddard Earth Sciences Data and Information Services Center for

OMI        (https://disc.gsfc.nasa.gov/datasets/OMNO2_003/summary)        and        TROPOMI        version        1 (https://disc.gsfc.nasa.gov/datasets/S5P_L2__NO2____1/summary). TROPOMI retrievals reprocessed to version 2.3.1 are available through the Sentinel-5P data portal (https://data-portal.s5p-pal.com/). WOUDC ozonesonde data, including data at the        Trinidad        Head,        California        launch        site,        are        available        through        WOUDC        at (https://woudc.org/data/dataset_info.php?id=ozonesonde). Hourly AQS $O_3$ and $NO_2$ observations are available from EPA's

Air Data website (https://aqs.epa.gov/aqsweb/airdata/download_files.html) (U.S. EPA, 2022b). GSI code is available via https://dtcenter.org/community-code/gridpoint-statistical-interpolation-gsi/download (DTC, 2018) CMAQ source code is freely available via https://github.com/usepa/cmaq.git and via the EPA (U.S. EPA, 2020).



*Author contributions.* JDE, BHH, SLN, SNK, and FGM designed the study. RBP, BHH, and DQT provided initial
conceptualization for the research. JDE conducted the air quality simulations, data assimilation, emissions inversions, and
data analysis. AL developed the OMI and TROPOMI observation operators in GSI. BHH and JDE developed the CMAQ
and GSI coupling. RG performed the meteorological modeling. GS implemented the halogen chemistry in the CMAQ
model. JDE, BHH, SLN, SNK, RBP, and FGM participated in data analysis and discussions. JDE wrote the paper with input
from all co-authors.


*Competing interests.* The authors declare that they have no conflict of interest.

*Disclaimer.* The views expressed in this article are those of the authors and do not necessarily represent the views or policies
of the U.S. Environmental Protection Agency.


*Acknowledgments.* J. D. East was supported, in part, by an appointment to the Research Participation Program at the Office
of Research and Development, U.S. Environmental Protection Agency, administered by the Oak Ridge Institute for Science
and Education through an interagency agreement between the U.S. Department of Energy and the EPA. D. Q. Tong and R.
B. Pierce acknowledge financial support from the NASA Health and Air Quality Applied Sciences Team (HAQAST) Tiger
Team under award number 80NSSC21K0427. We gratefully acknowledge the free availability and use of observational data
sets from AQS and WOUDC; remote sensing retrievals from OMI and TROPOMI; and global emission inventories from
CAMS, TCR-2, and GEIA. Comments by Heather Simon and Tanya Spero at the U.S. EPA served to strengthen this
manuscript.

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
