# Peer review of "Inferring and evaluating satellite-based constraints on NOx emissions estimates in air quality simulations"

_Atmospheric Chemistry and Physics, 2022_

## Author Response (AR1)

Response to Reviewers for manuscript 'Inferring and evaluating satellite-based constraints on NOx emissions estimates in air quality simulations' (https://doi.org/10.5194/acp-2022-435)

We would like to thank the reviewers for their valuable comments. Below we address each of the reviewers' comments. Author responses are in blue. Line numbers refer to the track-changes version of the manuscript.

**Referee #1** (https://doi.org/10.5194/acp-2022-435-RC1)

This manuscript performed NOx emission inversions using finite difference mass balance and OMI and TROPOMI data. The authors found systematic low biases in the estimates from TROPOMI due to the biases in the retrievals. They also found improvement in the posterior simulation of ozone with measurements. The topic fits the readership of ACP. However, several clarifications and more detailed discussions are needed, see details below.

*We thank reviewer 1 for the helpful comments and questions. A response to each comment is provided below.*

Specific Comments

Abstract, please include what is the period and spatial scale for this work.

*The abstract now states "We apply the framework to separately estimate lightning and anthropogenic NOx emissions over the Northern Hemisphere for 2019."*

L15-16, Please explain how 3D-Var plays a role in the inversion.

*The abstract now says "Here, we introduce an inverse modeling framework that couples satellite chemical data assimilation to a chemical transport model. In the framework, satellite-constrained emissions totals are inferred using model simulations with and without data assimilation in the iterative finite-difference mass-balance method."*

L73-74, it is not clear to me how this work is different from previous assimilations in addressing near-surface NO2 just based on this statement. Please elaborate more.

*In previous finite-difference mass-balance (FDMB) inversions, the $NO_2$ vertical column density (VCD) retrieved by the satellite instrument is directly compared to the model-simulated VCD. Emissions are therefore adjusted based on the difference between the full tropospheric column amounts, and not based on $NO_2$ near the surface, since this cannot be determined with the satellite VCD. In our framework, updates to model $NO_2$ concentrations by 3D-variational (3DVAR) data assimilation are vertically allocated to the model layers in the 3DVAR assimilation. As a result, it is possible to compute the partial VCD using only the model layers near the surface. By using the partial VCD of model simulations with and without data assimilation as inputs to the FDMB inversion, we target the near-surface $NO_2$ in a way that previous mass-balance based inversion did not. Lines 78-83 now read:*

*"In contrast to traditional FDMB, which directly compares modeled and observed columns, our framework improves the FDMB method by first assimilating satellite-retrieved $NO_2$, and then performing the inversion by comparing model simulations with and without assimilation. In the assimilation step, updates to model concentrations are vertically allocated to model layers. As a result, assimilating the*

*observed column allows the inversion framework to use only the near-surface portion of the model column in the FDMB inversion, minimizing influences from the upper troposphere and extending the framework proposed by Lamsal et al. (2011)."*

L197, "assimilating" causes some confusion here. A more proper word here might be "performing an inversion"?

In this sentence, "assimilating" best describes the method. However, we have rephrased the sentence to avoid the confusion. Lines 208-210 now read:

*"We modify the approach by first updating the model concentrations with assimilation of satellite observations, and then updating the emissions using the difference between the modeled VCD with and without assimilated satellite information."*

L224, please clarify whether you are using the total NO2 column to adjust lightning NOx or only NO2 observation in the upper atmosphere. It makes more sense to me if it is the latter.

When updating lightning-NOx emissions, we use the full tropospheric VCD from model simulations with data assimilation and model simulations without data assimilation as input to the FDMB. The model simulations that assimilate satellite data use a background error covariance that is created with a lightning-NO perturbed simulation. As a result, the assimilation is most sensitive to $NO_2$ in the upper troposphere where lightning NO emissions occur. In this way, we rely on the background error covariance to focus the lightning emissions on the upper troposphere. We have clarified this in the manuscript and lines 241-243 now read:

*"We then assimilate satellite $NO_2$ observations using the background errors for the upper troposphere and apply $\beta_{LNOx}$ in a single inversion iteration using the full tropospheric VCD to compute spatially-varying LNOx adjustment factors. Updates to LNOx are calculated using monthly averages."*

L255, again, I think most previous emission inversions did the same. It is not clear to me how the method used in this work differs from previous work.

The key difference is that in our framework we compare two modeled columns, one with assimilation and one without. In previous mass-balance inversions, model columns were directly compared to satellite columns. In that approach, it is not possible to compute a partial column because the satellite product is used directly. In our approach, we compute partial columns using the near-surface model layers, and use these partial near-surface columns as input to the FDMB inversion.

Figure 6, for the emission adjustments, could you discuss a bit more on how you determine whether it is due to emissions or chemistry in the model? Over those regions that have different seasonal adjustments in emissions, would that be due to different beta under different chemistry regimes? Otherwise, would the activity or emission factor of anthropogenic emissions have seasonal variations that are not well-captured?

In our approach and in all mass balance approaches, the difference between the VCDs ($\Delta\Omega$) is fully attributed to emissions. In reality, some of the difference may be due to other factors including chemistry and transport. The extent to which this introduces error to the inversion is an uncertainty in all mass balance methods. To highlight this uncertainty, we edited the text in lines 577-579:

*"As in all mass-balance based approaches, our method fully attributes the change in the VCD to emissions changes. To the extent that column differences are due to chemistry or transport and not*

*emissions, this assumption introduces error into mass-balance inversions, including the inversion implemented in our framework.”*

In regions where the sign of the emissions adjustment changes from season to season, the change in sign is due only to the change in the sign of $\Delta\Omega$, the VCD difference between the simulations with and without data assimilation. $\beta$ is positive by definition. Because our method assumes that all of the difference is due to emissions, seasonal differences indicate seasonal variations in activity or emission factors that are not well captured. In fact, identifying seasonal variations not previously captured in emissions inventories, particularly in regions with high emissions uncertainty, is a goal of this framework.

Figure 7, are the differences in emission adjustments proportional to the differences in the NO2 column from OMI and TROPOMI?

Yes, differences in emissions inferred by OMI or TROPOMI are proportional to differences between the OMI and TROPOMI $NO_2$ columns. The retrieved $NO_2$ columns are first assimilated to update the model $NO_2$ concentrations. Then, the emissions updates are calculated using the value of beta, the model $NO_2$ VCD without assimilation, and the model $NO_2$ VCD with assimilation. The model column with OMI $NO_2$ data assimilated and with TROPOMI $NO_2$ assimilated are both related to their respective emissions adjustments by the value of beta. As a result, the differences between these columns are proportional to differences between the emissions inferred by each satellite by definition.

L475, did you see better agreement in upper tropospheric NO2 between the simulation and observations after adjusting lightning NOx?

Figure 9 shows that both OMI and TROPOMI inferred updates to lightning NOx emissions improve model simulations compared to OMI and TROPOMI $NO_2$ satellite retrievals, respectively. In our model simulations, upper troposphere $NO_2$ concentrations increase with lightning NOx emissions updates from either satellite. In addition, Figures 10 and 11 show model comparisons to ozonesondes. We found that updates to lightning NOx emissions decreased bias in the upper troposphere compared to ozonesondes for OMI and TROPOMI updates. Increasing lightning NOx emissions led to increased modeled upper troposphere $NO_2$, improved model performance against column $NO_2$ satellite retrievals, and improved model performance against ozonesonde observations. This information, along with results of other studies which we cite in the manuscript showing that models tend to have a low bias for $NO_2$ in the upper troposphere (Goldberg et al., 2017; Qu et al., 2021; Silvern et al., 2019), strongly suggest that our lightning NOx emissions updates lead to better representation of upper troposphere $NO_2$.

Section 4, I presume the motivation to use this FDMB is to reduce computational cost. Please discuss how the computational cost of this work is compared to other methods.

The computational cost of this framework is greater than traditional FDMB because of the satellite data assimilation step that is not part of traditional FDMB. However, the computational cost of our approach is similar or less than other top-down methods such as Kalman Filter and adjoint 4D-variational based approaches. We have added discussion on lines 569-573:

*“The computational cost is greater than that of traditional FDMB inversions due to the assimilation step. However, the computational burden is comparable or less than other satellite assimilation methods such as Kalman-Filter and adjoint 4D-variational approaches. In addition, the framework requires minimal code changes to the underlying CTM, so inverse estimates will improve as the underlying air quality model is updated with little additional effort needed to implement this framework.”*

**References**

Goldberg, D. L., Anenberg, S. C., Lu, Z. F., Streets, D. G., Lamsal, L. N., McDuffie, E. E., and Smith, S. J.: Urban NOx emissions around the world declined faster than anticipated between 2005 and 2019, Environ Res Lett, 16, 10.1088/1748-9326/ac2c34, 2021.

Lamsal, L. N., Martin, R. V., Padmanabhan, A., van Donkelaar, A., Zhang, Q., Sioris, C. E., Chance, K., Kurosu, T. P., and Newchurch, M. J.: Application of satellite observations for timely updates to global anthropogenic NOx emission inventories, Geophys Res Lett, 38, 10.1029/2010gl046476, 2011.

Qu, Z., Jacob, D. J., Silvern, R. F., Shah, V., Campbell, P. C., Valin, L. C., and Murray, L. T.: US COVID-19 Shutdown Demonstrates Importance of Background NO2 in Inferring NOx Emissions From Satellite NO2 Observations, Geophys Res Lett, 48, 10.1029/2021GL092783, 2021.

Silvern, R. F., Jacob, D. J., Mickley, L. J., Sulprizio, M. P., Travis, K. R., Marais, E. A., Cohen, R. C., Laughner, J. L., Choi, S., Joiner, J., and Lamsal, L. N.: Using satellite observations of tropospheric NO2 columns to infer long-term trends in US NOx emissions: the importance of accounting for the free tropospheric NO2 background, Atmos Chem Phys, 19, 8863-8878, 10.5194/acp-19-8863-2019, 2019.

**Referee #2** (https://doi.org/10.5194/acp-2022-435-RC2)

The authors developed an inverse modeling framework to infer NOx emissions by assimilating satellite observations. The highlight of the method is to separate the surface and lightning emissions contributions. However, I don't fully understand how the separation has been achieved by reading section 2.5. Additionally, the writing may need improvement. I recommend careful copy editing of the manuscript.

We thank reviewer 2 for the helpful comments and questions. A response to each comment is provided below. The changes made in response to the reviewer's comments have improved the manuscript and made it clearer.

General comments:

CMAQ model does not include stratosphere simulations. Will this contribute to uncertainties in the inferred emissions? I recommend adding some discussion about this.

In our framework, we take several steps to ensure that only the troposphere is considered in the inversion. First, in the data assimilation, the model top is cut off at the tropopause pressure reported by the satellite retrieval. Second, we only apply the inversion in grid cells which are dominated by anthropogenic emissions. As a result, background areas where the stratosphere contributes the most uncertainty to model-satellite comparisons are not considered, and uncertainties in the inversion due to the lack of stratospheric chemistry are minimized. However, as the reviewer suggests, this is an uncertainty, and lines 585-588 discuss this:

*"The air quality model used here does not include stratospheric chemistry, which could affect comparisons against $NO_2$ retrievals. Nevertheless, the framework shows the potential to improve air quality model predictions using satellite-derived emissions updates, in particular for regions with highly uncertain emissions inventories."*

Section 2.1. Fig 2. The OMI/TROPOMI ratio is larger than 2 for large areas. I fully understand that it is time-consuming to integrate the TROPOMI data into the system and thus the authors did not update it. But the tremendous differences between those two datasets as shown in Fig 2 make me worried about the uncertainties associated with the usage of the early version of TROPOMI data. I suggest at least running the system with updated TROPOMI data for a short period, like a month, to have an understanding of the uncertainties.

In figure 2, there are large areas where the OMI/TROPOMI ratio is in the range "1.25-2.00" but very few grid cells in which the ratio is greater than 2. Nevertheless, we thank the reviewer for pointing this out and we agree that it is an important uncertainty. We included a 1-month simulation and inversion with updated v2.3.1 TROPOMI data in the original manuscript. The results are presented in Figures S10 and S11 and show an increase in inferred NOx emissions during that time compared to previous version of TROPOMI $NO_2$. Lines 423-426 in the original text read:

*"We conduct an inversion using the reprocessed TROPOMI $NO_2$ version 2.3.1 (Van Geffen et al., 2021) to infer NOx emissions for January 2019, and find that the updated data increases the TROPOMI posterior inference by 17% over the U.S. and 4% in China relative to version 1.2.2, but still differs significantly from that obtained using OMI observations (Figs. S10 and S11)."*

Section 2.4. Lamsal et al use OMI NO$_2$ to calculate delta omega and omega. In this work, the authors propose to use CMAQ simulations for the calculation alternatively. What is the advantage of using CMAQ simulations compared to satellite NO2 observations?

In the finite-difference mass-balance (FDMB) inversion introduced by Lamsal et al. (2011), the NO$_2$ vertical column density (VCD) retrieved by the satellite instrument is directly compared to the model-simulated VCD. The key difference in our framework is that we compare two modeled columns, one with assimilation and one without. In the original approach, it is not possible to compute a partial column because the satellite product is used directly. In our approach, using the model columns with and without assimilated NO$_2$ retrievals allows us to compute partial columns using the near-surface model layers, and then use these partial near-surface columns as input to the FDMB inversion. This has the advantage of eliminating differences that occur in the upper troposphere from the inversion and focusing the inversion on the part of the column that is dominated by anthropogenic emissions. Figure S1 demonstrates potential pitfalls that are avoided by using the only the bottom portion of the column in this way.

To increase clarity, lines 78-83 now read:

*"In contrast to traditional FDMB, which directly compares modeled and observed columns, our framework improves the FDMB method by first assimilating satellite-retrieved NO$_2$, and then performing the inversion by comparing model simulations with and without assimilation. In the assimilation step, updates to model concentrations are vertically allocated to model layers. As a result, assimilating the observed column allows the inversion framework to use only the near-surface portion of the model column in the FDMB inversion, minimizing influences from the upper troposphere and extending the framework proposed by Lamsal et al. (2011)."*

Section 2.5. Do the authors treat all changes in NO2 as lightning emissions changes for non-populated areas? Do anthropogenic emissions only cover populated areas? If so, I would recommend clarifying and pointing this out in the abstract and conclusion.

Lightning NOx and anthropogenic NOx updates are performed in two steps and are updated separately. Anthropogenic emissions updates are only applied over populated areas, while no such restriction is considered for lightning emissions. The restrictions applied to anthropogenic emissions are described in Section 2.5. In addition, we have made the following edits.

The abstract now states: *"We apply the framework to separately estimate lightning and anthropogenic NOx emissions over the Northern Hemisphere for 2019"*

Line 236 now says:

*"In our framework, LNOx emissions are updated first, separately from anthropogenic emissions"*

Lines 244-245 in Section 2.5 now say:

*"After LNOx emissions are updated, ANOx emissions are updated by iteratively applying a FDMB inversion independently for each month in 2019."*

Lines 557-559 in the conclusions now say:

*"Here, we implement the framework in a NOx emissions inversion to separately update 2019 Northern Hemisphere lightning and anthropogenic NOx emissions estimates using NO$_2$ products from the OMI and TROPOMI satellite instruments."*

Is there any specific reason to use both OMI and TROPOMI? It seems the system works well by using a single instrument. I recommend clarifying the pro of using two instruments.

The emissions are updated by either OMI, or TROPOMI, and are not updated by both satellite instruments simultaneously. Two emissions datasets are produced as a result – one updated by OMI, and one updated by TROPOMI. Since both instruments are commonly used in research to update NOx emissions, a purpose of this paper is to compare emissions using either instrument. This is described in the introduction and methods. In addition, we have modified lines 83-84 for greater clarity:

*"In addition, our analysis compares independent inversions which separately use OMI or TROPOMI $NO_2$ data."*

Specific comments:

line 20. Inferred from?

The abstract now reads: *"Using overlapping observations from the Ozone Monitoring Instrument (OMI) and the Tropospheric Monitoring Instrument (TROPOMI), we compare separate NOx emissions inferences from these satellite instruments, as well as the impacts of emissions changes on modeled $NO_2$ and $O_3$."*

Line 22. Smaller bias?

In this case, we intentionally use the term "low bias" to refer to the known low bias in early versions of the TROPOMI $NO_2$ retrievals. TROPOMI $NO_2$ demonstrates a larger bias than OMI $NO_2$.

Line 24. Improve performance or reduce bias?

Our results show improved model performance against observations with lightning NOx updates.

line 25. This sentence is the conclusion sentence of the abstract, but it looks lengthy and unclear to me. I recommend rephasing this sentence substantially.

The abstract now says "*The combined lighting and anthropogenic emissions updates improve the model's ability to reproduce measured ozone by adjusting natural, long-range, and local pollution contributions. Thus, the framework informs and supports the design of domestic and international control strategies.*"

Line 33. It is more common to use "global inventory".

We have made this change.

Line 42. Why is the bottom-up inventory incomplete?

Despite the large amount of detailed work that goes into creating bottom-up inventories, there are inevitably uncertainties in the inventories, and it is often not feasible to update global inventories in real time. This is a strength of top-down inversions, which can update emissions inventories in real time based on observations, although top-down inversions come with their own set of uncertainties. In the manuscript, we briefly describe these factors while giving credit to the valuable time and effort that goes into creating bottom-up inventories. We also point to three papers that explore, in detail, the uncertainties in bottom-up inventories (McDuffie et al., 2020; Elguindi et al., 2020; Day et al., 2019).

Line 42-43. Are you indicating that large uncertainties in emissions estimates for developing countries will propagate into that for developed countries? Please try to rephrase the sentence here. The current meaning is unclear to me.

Lines 46-48 now read: *"Uncertainties in bottom-up emissions estimates are particularly large for high income countries (HICs) (McDuffie et al., 2020; Elguindi et al., 2020) and remain significant for developed countries (Day et al., 2019)."*

Line 48. What is the definition of detailed emissions updates?

In this case, we mean "precise" emissions updates. Lines 52-53 now read: *"Adjoint-based methods can provide precise emissions updates, but require significant computational resources"*

Line 57. What is the emissions smearing effect?

Lines 62-63 now read: *"the method is subject to an emissions smearing effect (e.g. Cooper et al., 2017), which can cause emissions updates to be spatially misallocated"*

Line 59. Do the authors indicate that averaging a few observations will reduce the biases significantly? I assume the averaging will help to reduce noise, but not systematic bias.

We do are not attempting to make this claim. Rather, the sentence is referring to the value of comparing independent inversions with different satellite instruments.

Line 73. Please try to briefly explain the reason why assimilation allows for minimizing influence from the upper troposphere before claiming it. It is also not clear to me how it will be an extension of the work of Lamsal et al.

Lines 78-83 now read: *"In contrast to traditional FDMB, which directly compares modeled and observed columns, our framework improves the FDMB method by first assimilating satellite-retrieved NO$_2$, and then performing the inversion by comparing model simulations with and without assimilation. In the assimilation step, updates to model concentrations are vertically allocated to model layers. As a result, assimilating the observed column allows the inversion framework to use only the near-surface portion of the model column in the FDMB inversion, minimizing influences from the upper troposphere and extending the framework proposed by Lamsal et al. (2011)."*

We also refer the reviewer to our response to the third general comment in this review regarding how this work differs from and extends that of Lamsal et al.

Line 83. I rarely see the term "surface-based observations". I suggest a more common-used term of ground observations.

We have changed this to term to "ground-based observations".

Fig 1. Caption. I don't quite get the meaning of "the boundary of the inversion algorithm".

The caption has been changed and now reads: *"Red dotted lines around the inversion boxes correspond to the red dotted lines in Fig. 3, which details the inversion algorithm."*

Line 118. Improvement of the bias?

The sentence is referring to the known low bias in TROPOMI NO$_2$ described in the previous sentence.

Line 122. Results should not be capitalized.

We have made the change.

Line 138. What is representative-day?

Lines 148-150 now have additional description: *"For each day of the week and each month, there is a unique hourly emissions file that is used for every matching day of the week in that month. As a result, diurnal and weekly patterns are captured in the emissions, while daily variations that are specific to the prior emissions inventory year are averaged."*

Sect 2.2. What is the spatial resolution of the CMAQ simulation?

CMAQ simulations are run at 108km horizontal resolution with 44 vertical layers up to 50 hPa. This information is included in Section 2.2.

References

Cooper, M., Martin, R. V., Padmanabhan, A., and Henze, D. K.: Comparing mass balance and adjoint methods for inverse modeling of nitrogen dioxide columns for global nitrogen oxide emissions, J Geophys Res-Atmos, 122, 4718-4734, 10.1002/2016jd025985, 2017.

Day, M., Pouliot, G., Hunt, S., Baker, K. R., Beardsley, M., Frost, G., Mobley, D., Simon, H., Henderson, B. B., Yelverton, T., and Rao, V.: Reflecting on progress since the 2005 NARSTO emissions inventory report, J Air Waste Manage, 69, 1023-1048, 10.1080/10962247.2019.1629363, 2019.

Elguindi, N., Granier, C., Stavrakou, T., Darras, S., Bauwens, M., Cao, H., Chen, C., van der Gon, H. A. C. D., Dubovik, O., Fu, T. M., Henze, D. K., Jiang, Z., Keita, S., Kuenen, J. J. P., Kurokawa, J., Liousse, C., Miyazaki, K., Muller, J. F., Qu, Z., Solmon, F., and Zheng, B.: Intercomparison of Magnitudes and Trends in Anthropogenic Surface Emissions From Bottom-Up Inventories, Top-Down Estimates, and Emission Scenarios, Earths Future, 8, 10.1029/2020EF001520, 2020.

Lamsal, L. N., Martin, R. V., Padmanabhan, A., van Donkelaar, A., Zhang, Q., Sioris, C. E., Chance, K., Kurosu, T. P., and Newchurch, M. J.: Application of satellite observations for timely updates to global anthropogenic NOx emission inventories, Geophys Res Lett, 38, 10.1029/2010gl046476, 2011.

McDuffie, E. E., Smith, S. J., O'Rourke, P., Tibrewal, K., Venkataraman, C., Marais, E. A., Zheng, B., Crippa, M., Brauer, M., and Martin, R. V.: A global anthropogenic emission inventory of atmospheric pollutants from sector- and fuel-specific sources (1970-2017): an application of the Community Emissions Data System (CEDS), Earth Syst Sci Data, 12, 3413-3442, 10.5194/essd-12-3413-2020, 2020.

van Geffen, J., Eskes, H., Compernolle, S., Pinardi, G., Verhoelst, T., Lambert, J. C., Sneep, M., Ter Linden, M., Ludewig, A., Boersma, F., and Veefkind, J. P.: Sentinel-5P TROPOMI NO2 retrieval: impact of version v2.2 improvements and comparisons with OMI and ground-based data, Atmospheric Measurement Techniques Discussions, [preprint]. In review., 10.5194/amt-2021-329, 2021.

---

## Author Response (AR2)

Response to Reviewers for manuscript 'Inferring and evaluating satellite-based constraints on NOx emissions estimates in air quality simulations' (https://doi.org/10.5194/acp-2022-435)

We would like to thank the reviewers for their valuable comments. Below we address each of the reviewers' comments. Author responses are in blue. Line numbers refer to the track-changes version of the manuscript.

Referee comments

Lines 585-588: Nevertheless, the framework shows the potential to improve air quality model predictions using satellite-derived emissions updates, in particular for regions with highly uncertain emissions inventories. Are the authors suggesting that the uncertainties of emissions for polluted regions are more uncertain than in other regions here? I'm not sure about it since the uncertainty of natural emissions is large as well. Please cite a solid reference for such a claim.

We have modified the statement to be more specific and to include a reference to Elguindi et al. (2020). In that study, authors demonstrate that in regions where emissions factors and activity information are uncertain and in regions where emissions are rapidly changing, top-down satellite inferences provide an opportunity to complement uncertain bottom-up inventories and improve emissions estimates. Our study further demonstrates an improvement in air quality simulations with top-down emissions inferences.

*Lines 573-574 now read: Nevertheless, the framework shows the potential to improve air quality model predictions using satellite-derived emissions updates, in particular for regions with uncertain emissions inventories or undergoing rapid emissions changes (Elguindi et al., 2020).*

Uncertainty about using the early version of TROPOMI data. Thanks for including the additional 1-month assimilation. You may want to point out that the impact of changing the date version is limited if that is the case.

We have enhanced our discussion of the additional 1-month assimilation with updated TROPOMI retrievals. Lines 411-416 now read: *We conduct an inversion using the reprocessed TROPOMI $NO_2$ version 2.3.1 (Van Geffen et al., 2021) to infer $NO_x$ emissions for January 2019, and find that the updated data increases the TROPOMI posterior inference by 17% over the U.S. and 4% in China relative to version 1.2.2. While using the updated retrievals shrinks the gap between OMI and TROPOMI inferred emissions, it does not change the overall trend of smaller posterior emissions using TROPOMI $NO_2$ (Figs. S10 and S11).*

References

Elguindi, N., Granier, C., Stavrakou, T., Darras, S., Bauwens, M., Cao, H., Chen, C., van der Gon, H. A. C. D., Dubovik, O., Fu, T. M., Henze, D. K., Jiang, Z., Keita, S., Kuenen, J. J. P., Kurokawa, J., Liousse, C., Miyazaki, K., Muller, J. F., Qu, Z., Solmon, F., and Zheng, B.: Intercomparison of Magnitudes and Trends in Anthropogenic Surface Emissions From Bottom-Up Inventories, Top-Down Estimates, and Emission Scenarios, Earths Future, 8, 10.1029/2020EF001520, 2020.